# The role of mountains in shaping the global meridional overturning circulation

Haijun Yang [1] ✉, Rui Jiang[2], Qin Wen[3], Yimin Liu [4,5], Guoxiong Wu[4,5] & Jianping Huang [6]

The meridional overturning circulation (MOC) in the ocean is a key player in the global climate system, while continental topography provides an essential backdrop to the system. In this study, we design a series of coupled model sensitivity experiments to investigate the influence of various mountain ranges on the global thermohaline circulation. The results highlight the influence of the Tibetan Plateau (TP) on the global thermohaline circulation. It emerges as a requisite for establishing the Atlantic MOC (AMOC) and a determining factor for the cessation of the Pacific MOC (PMOC). Additionally, the Antarctic continent plays a vital role in facilitating the TP to form the AMOC. While the formation of the AMOC cannot be attributed to any single mountain range, the TP alone can inhibit the PMOC's development. By modifying the global hydrological cycle, the TP is likely to have been crucial in molding the global thermohaline circulation.

The meridional overturning circulation (MOC) in the ocean plays a pivotal role in both regional and global climate, via large-scale heat and freshwater transports. In the current climate, the MOC exists mainly in the Atlantic, referred to as the Atlantic MOC (AMOC), which is characterized by deep convection and deep-water formation in the Labrador Sea and Greenland-Iceland-Nordic (GIN) seas of the subpolar Atlantic[1,2]. The observed strength of the AMOC is approximately 18 Sv[3]. There is no equivalent Pacific MOC (PMOC) in the North Pacific.

Geological evidence reveals that the primary deep-water formation region in the Northern Hemisphere (NH) might have undergone a shift from the Pacific to the Atlantic in the past. Some earlier studies suggested that North Pacific deep-water (NPDW) formation was strong during the Paleocene period (about 65–55 million years ago, or Ma)[4], while the North Atlantic deep-water (NADW) formation was weak and likely began to develop at a later stage[5,6]. However, a recent study from the DeepMIP project found that neither model results nor proxy data suggest NADW formation during the early Eocene, while evidence for NPDW formation remains elusive. Consequently, the modern AMOC

might initially develop in the late Miocene (about 12–9 Ma) and not be fully established until the late Pliocene to early Pleistocene (about 4–3 Ma)[7]. Nonetheless, a recent study from DeepMIP project found that neither model results nor proxy data suggest NADW formation during the early Eocene, while the evidence for NPDW formation remains inconclusive[8]. The actual evolutionary history of the AMOC and PMOC therefore remains a topic of considerable debate.

The asymmetry of net surface freshwater flux is often cited as the cause of different overturning modes between the Atlantic and Pacific. The North Atlantic has higher sea-surface salinity (SSS) than the North Pacific because the former is a net evaporation basin, while the latter is nearly neutral[9]. Additionally, ocean basin geometry plays a role in the different overturning modes. Research indicated that narrow basins are more conducive to deep overturning circulation than wide basins[2,10,11]. Furthermore, ocean gateways also contribute to the different overturning modes. The opening of the Drake Passage/Tasman Seaway in the late Eocene is thought to have promoted the NADW formation and thus the AMOC formation[12].

[1]Department of Atmospheric and Oceanic Sciences and Key Laboratory of Polar Atmosphere-ocean-ice System for Weather and Climate of Ministry of Education, Fudan University, Shanghai 200438, China. [2]Department of Atmospheric and Oceanic Sciences, School of Physics, Peking University, 100871 Beijing, China. [3]School of Geography, Nanjing Normal University, Nanjing 210023, China. [4]State Key Laboratory of Numerical Modelling for Atmospheric Sciences and Geophysical Fluid Dynamics, Institute of Atmospheric Physics, Chinese Academy of Sciences, 100029 Beijing, China. [5]College of Earth Science, University of Chinese Academy of Sciences, 100049 Beijing, China. [6]Collaborative Innovation Center for Western Ecological Safety, College of Atmospheric Sciences, Lanzhou University, Lanzhou 730000, China. ✉e-mail: yanghj@fudan.edu.cn

Geological evidence also suggests that the uplift of large continental mountains has had a significant impact on the climate[13]. The evolution of continental terrain holds the potential to trigger large-scale transitions in the global MOC (GMOC)[14–16], and in the following, we briefly review the uplift periods of the major modern mountain ranges. The Rocky Mountains (RMs) rose from the sea level about 80 Ma[17], and reached its current elevation about 45 Ma[8]. Although the Transantarctic and Gamburtsev Mountains over Antarctica were likely already present at the start of the Cenozoic (65 Ma)[18,19], the first large-scale glaciation of Antarctica is believed to have occurred during the Eocene-Oligocene transition period (34–33.5 Ma)[20,21]. The uplift of Andes Mountains (AMs) started in the Late Cretaceous (~70 Ma)[22] and matured around 15–10 Ma[23,24]. The uplift of these mountains predated the onset of the NADW formation. The Greenland (GL) underwent its initial phase of uplift in the late Miocene (11–10 Ma)[25]. The timeline for the formation of the Tibetan Plateau (TP) is a topic of highly debate. Some studies argue that parts of the TP were already in place during the late Eocene (38–33 Ma)[26,27], while other research suggest that the TP's rapid and main uplift occurred between 10 and 8 Ma[28–31]. A more recent study proposes that most of the TP had attained its current elevation before the Mid-Miocene (15 Ma)[32]. This timing coincides with the onset of NADW formation, suggesting a possible link between the TP uplift and the development of NADW. Recent research also indicates that the TP is a critical factor affecting changes in the GMOC[33,34]. Nevertheless, it is important to recognize that the chronology of the uplift of major mountain ranges remains a subject of intense discussion and investigation.

Here, as a first step, we numerically investigate how the presence or absence of various major mountain ranges affect the GMOC as well as their cumulative impact when they are uplifted sequentially in the model from an initial flat Earth. In order to isolate the topographic effects from plate tectonics and long-term bathymetric and atmospheric changes, we keep the modern bathymetry and continental positions, as well as modern greenhouse gas concentrations, incident solar radiation and orbital parameters. We show that, in our simulations (Methods and Table 1), the TP uplift is the primary driver of the PMOC shutdown and the AMOC initiation, although high Antarctic topography is required to drive a strong, modern-like AMOC. We further discuss the implications of our results to the long-term Cenozoic history of the AMOC/PMOC.

## Results

Adding mountains sequentially to flat continent in the model can ultimately lead to the establishment of the AMOC and the collapse of the PMOC (Fig. 1). However, this dramatic change occurs only after the TP is added. In the Flat scenario, the MOC features a modest PMOC of about 109 Sv and a negligible AMOC of less than 1 Sv, consistent with previous findings[14–16,34,35]. Following the TP uplift in model year 4001 (red curve), the AMOC recovers rapidly, overshoots its realistic value in about 800 years, and reaches an equilibrium state in about 1000 years, which matches the normal state in Real of about 18 Sv (gray line). Overshooting during AMOC recovery is common in freshwater hosing experiments[36], $CO_2$ forcing experiments[37], and other paleoclimate simulations[38], although their mechanisms may differ. Simultaneously, the PMOC collapses quickly (blue curve), and reaches an "off" state in Real of about 4 Sv in about 200 years. It suggests that the AMOC would be difficult to establish in the absence of the TP, while the PMOC would not exist in the presence of the TP.

None of the individual topography uplift from the Flat terrain is capable of initiating the AMOC. However, the TP uplift alone can cause the collapse of the PMOC (Fig. 2a). The presence of individual AT and AM, on the other hand, can result in a stronger PMOC than that in Flat. As demonstrated by Jiang and Yang[39], the presence of the RM alone has negligible effects on both AMOC and PMOC. Although the presence of the TP alone leads to a slight increase in the AMOC strength (approximately 2 Sv), it is inadequate for its complete establishment. Conversely, the PMOC is practically abolished in the presence of the TP alone.

Studies have demonstrated that the removal of the TP can result in the collapse of the AMOC[33,34,40]. This implies that the combination of the RM, AM, AT, and GL is insufficient to sustain the AMOC. Indeed, this is precisely the situation before the TP is added, as illustrated in Fig. 1. Moreover, we investigated the impacts of various combinations of RM, AM, AT, and GL; and none of these arrangements can establish the AMOC, irrespective of the order in which they were introduced to Flat. As the TP alone cannot support the establishment of the AMOC, we are left with the question: what is the minimum topographic requirement, in addition to the presence of the TP, for the establishment of the AMOC?

We find that only the combination of the TP and AT can lead to the establishment of the AMOC from Flat, whereas the combination of the

## Table 1 | Description and annotation of experiments

| Experiment | Year | AMOC | PMOC | Description |
|---|---|---|---|---|
| Flat | 0001-1600 | 0.7 | **12.0** | Flat global topography (50 m) |
| AM | 0801-1600 | 0.5 | **15.8** | Add AM to Flat |
| AT | 0801-2400 | 1.1 | **16.6** | Add AT to Flat |
| GL | 0801-1600 | 0.4 | **12.5** | Add GL to Flat |
| RM | 0801-2000 | 0.4 | 7.1 | Add RM to Flat |
| TP | 0801-3000 | 2.4 | 2.0 | Add TP to Flat |
| RM + AT | 2001-2400 | 0.9 | **10.6** | Add AT to the previous stage |
| RM + AT + AM | 2401-3400 | 2.1 | **21.2** | Add AM to the previous stage |
| RM + AT + AM + GL | 3401-4000 | 1.5 | **21.9** | Add GL to the previous stage |
| RM + AT + AM + GL + TP | 4001-6000 | **18.1** | 3.5 | Add TP to the previous stage |
| TP + AM | 0801-3000 | 5.0 | 2.0 | Add TP and AM to Flat |
| TP + GL | 0801-3000 | 1.7 | 2.4 | Add TP and GL to Flat |
| TP + AT | 0801-3000 | **19.4** | 2.5 | Add TP and AT to Flat |
| TP + RM | 0801-3000 | 2.4 | 2.4 | Add TP and RM to Flat |
| Real | 0801-3000 | **18.2** | 3.6 | Add all topography simultaneously to Flat |

"RM, AT, AM, GL, and TP" represent the Rocky Mountains, the Antarctic, the Andes Mountains, Greenland, and the Tibetan Plateau, respectively. "All" represents the global topography. "Year" represents the integration length of the experiment. The strengths of Atlantic meridional overturning circulation (AMOC) and Pacific meridional overturning circulation (PMOC) (units: Sv) are obtained by averaging results over the last 100 years of the integration. Strong AMOC and PMOC are marked by bold face.

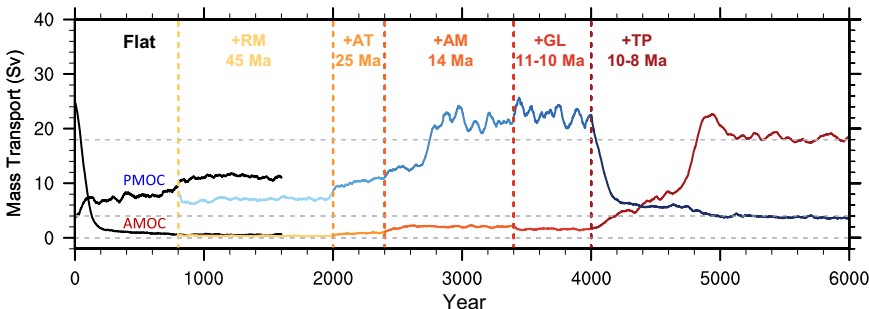

**Fig. 1 | Temporal evolutions of the Atlantic meridional overturing circulation (AMOC) and Pacific meridional overturning circulation (PMOC) with the sequential uplift of different mountains in Flat2Real.** The AMOC is represented by a red-scale colored curve, and the PMOC, by a blue-scale colored curve. Each segment corresponds to one mountain uplift. The "+" sign is used to indicate that a topography is added to the previous stage. The AMOC and PMOC indexes are defined as the maximum meridional overturning streamfunction (units: Sv, $1\,Sv = 10^6\,m^3/s$) between 20° and 70°N and below 500-m depth in the North Atlantic and North Pacific, respectively. Each time series is smoothed using a 51-year running mean. The mountain uplifts here are the Rocky Mountains (RM), Antarctic (AT), Andes Mountains (AM), Greenland (GL), and Tibetan Plateau (TP).

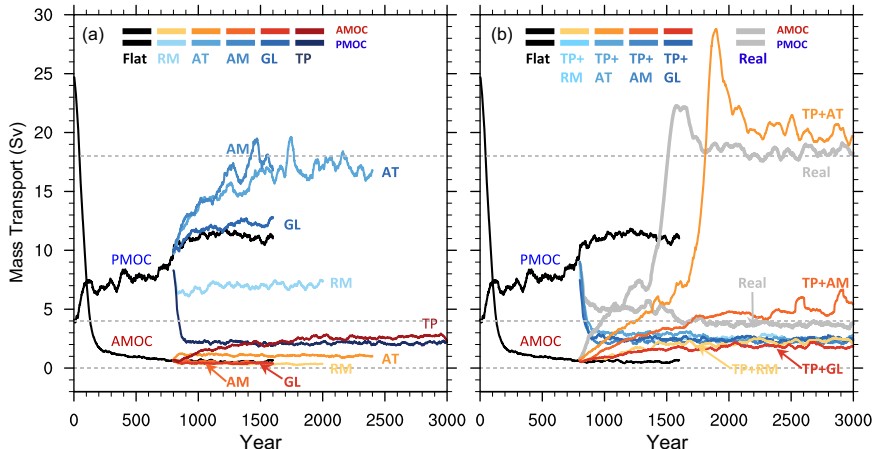

**Fig. 2 | Temporal evolution of the Atlantic meridional overturing circulation (AMOC) and Pacific meridional overturning circulation (PMOC) in different topography experiments.** Details of these experiments are given in Table 1. The AMOC and PMOC are represented by red-scale and blue-scale colored curves, respectively. **a** Results of individual topography experiments. **b** Results of Tibetan Plateau (TP) plus another topography. All curves are smoothed using a 51-year running mean. Dashed gray reference lines in **a** and **b** indicate the mean strengths of the AMOC and PMOC in Real, which are 18 and 4 Sv, respectively.

TP with any other mountains is ineffective (Fig. 2b). With TP + AT, the AMOC gradually increases within the first 800 years, followed by a sharp increase to a very high level in a short time, ultimately reaching quasi-equilibrium in approximately 2000 years, with the magnitude equivalent to that in Real. However, in the other combinations, even after an integration of 2200 years, the AMOC does not increase significantly. These experiments suggest that only the AT effectively assists the TP in the formation of the AMOC. Conversely, the PMOC quickly disappears in any experiment where the TP is present.

Figure 3 shows the patterns of the AMOC, PMOC, and GMOC in each experiment. In Flat (Fig. 3a1, a2), only the shallow wind-driven subtropical cells (STCs) are present in the Atlantic; while the PMOC is strong, it has a weak inter-hemispheric structure (Fig. 3b1, b2), consisting of the wind-driven STCs and subpolar thermohaline circulations. In the Pacific, the wind-driven STCs are strong and hemispherically symmetric, occupying the upper 500-m ocean between 30°S and 30°N. The thermohaline circulation originates in the North Pacific north of 30°N, and its lower branch occupies the deep ocean beneath 1000 m, extending to the Southern Ocean. The PMOC in the South Pacific (Fig. 3b2) is much weaker than the AMOC in the South Atlantic in Real (Fig. 3a8). The structure of strong PMOC and weak AMOC is maintained in Flat2Real before the TP uplift, and in the experiments without the TP.

The uplift of the TP appears to have fundamentally changed the GMOC. Based on the experiments we conducted, we conclude that the TP is a necessary condition for the "on" state of the AMOC and a sufficient condition for the "off" state of the PMOC. The AT is also critical to the establishment of the AMOC. Among the various topographic features, the TP stands the most significant factor influencing the GMOC.

Compared to Real, the MOC in Flat results from three key factors: 1) higher net evaporation in the North Pacific and greater precipitation in the North Atlantic, 2) stronger Ekman downwelling in the North Pacific, and 3) weaker Ekman pumping in the Southern Ocean and associated weaker southward water mass transport in the intermediate-to-deep South Pacific. The first two factors contribute to the formation of the NPDW and the shutdown of the NADW, while the third factor weakens the thermohaline component of the PMOC. The impacts of net evaporation in the North Atlantic and Ekman pumping in the Southern Ocean on the AMOC have been well recognized[41–46]. Here, we quantify their roles using topography experiments.

In Real, the sea-surface density (SSD) in the subpolar Atlantic is over 27.5 $\sigma_0$, which is sufficient for the NADW formation, even with Ekman upwelling (Fig. 4b). However, in Flat the SSD in the North Atlantic is reduced to 25.5 $\sigma_0$ (Fig. 4a); and this reduction ($\Delta\sigma \sim 2.0\,kg/m^3$) (Fig. 4e) is significant enough to halt the NADW formation. In Real, the

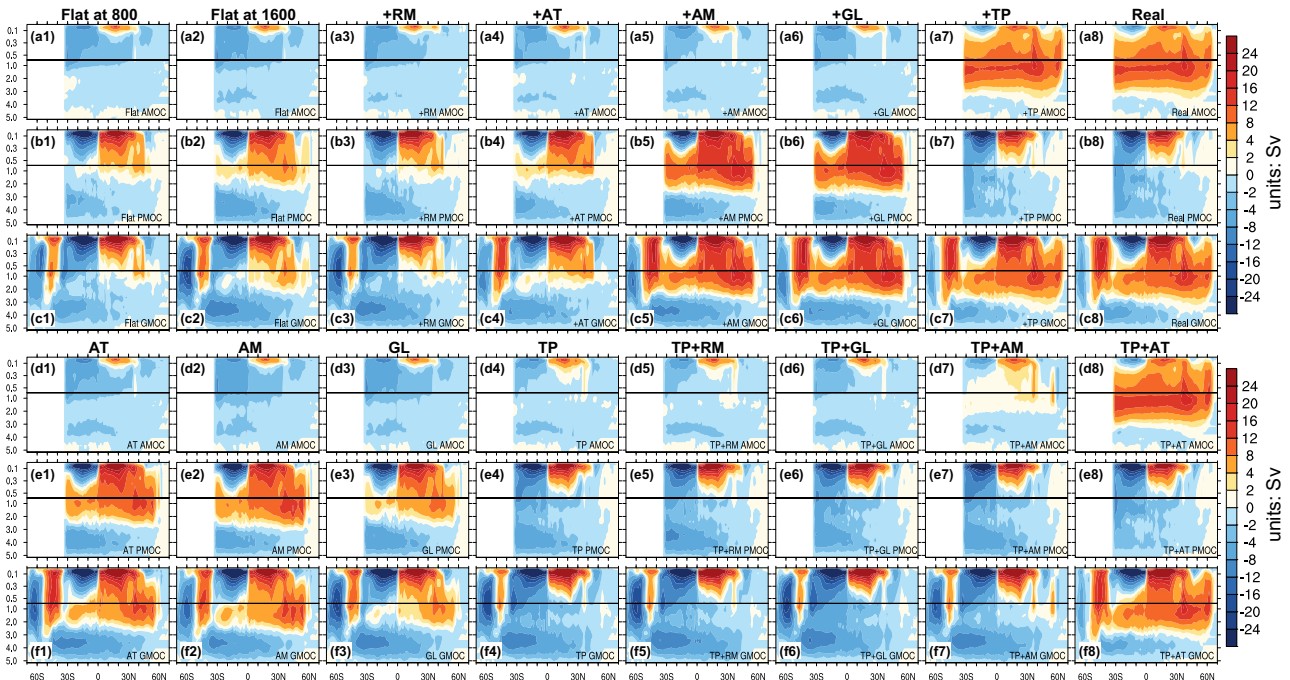

**Fig. 3 | Patterns of the meridional overturning circulation (MOC) in different experiments. a–c** The Atlantic MOC (AMOC) and Pacific MOC (PMOC), and global MOC (GMOC) in Flat2Real, respectively. The first (second) column shows the MOC patterns averaged over years 701–800 (years 1501–1600) of Flat. The third to seventh columns show the MOC patterns averaged over years 1801–2000, 2201–2400, 3201–3400, 3801–4000, and 5601–6000 of Flat2Real, respectively, corresponding to the sequential uplift of different mountains. The last column shows the MOC pattern averaged over years 2501–3000 of Real. **d–f** The MOC patterns for the last 200 years of each experiment (from left to right): AT (Antarctic), AM (Andes Mountains), GL (Greenland), TP (Tibetan Plateau), TP + RM (Rocky Mountains), TP + GL, TP + AM, and TP + AT. The vertical coordinate is depth (units: km).

North Pacific SSD is 25.5-26 $\sigma_0$; and in Flat, it is increased to 26.5 $\sigma_0$ in the Northeast Pacific (Fig. 4a). This increase ($\Delta\sigma$ ~ 0.5 kg/m³) is sufficient for shallow surface-water subduction, but is insufficient for deep-water formation, which requires the assistance from enhanced Ekman downwelling (Fig. 4e)[34]. Note that the physical processes involved in NADW formation in Real and NPDW formation in Flat are not the same. The former primarily involves thermohaline dynamics, while the latter involves both thermohaline and wind-driven dynamics[35].

The change in SSD from Real to Flat can be attributed to the change in SSS (Fig. 4i), in which the virtual salt flux resulting from net evaporation (i.e., evaporation minus precipitation, or EMP) and sea-ice melting (Supplementary Fig. 2b1) playing crucial roles. The contribution of sea-surface temperature change to SSD change is limited[34]. For a steady state, EMP across the ocean surface is almost equivalent to the vertically integrated water vapor transport divergence ($\nabla \cdot \vec{v} q$) throughout the entire atmosphere column, when freshwater flux from land surface and river runoff are disregarded.

In the transition from Real to Flat, aided by anomalous lows over the North Atlantic and North American continent (Supplementary Fig. 3b1), more atmospheric moisture is transported eastward from the North Pacific and north-eastward from the central tropical Pacific to the North Atlantic (Supplementary Fig. 3a1), leading to increased moisture convergence over the North Atlantic. Later, the southward expansion of sea ice in the subpolar Atlantic results in sea-ice melting at the same time, providing a substantial amount of fresh water into the ocean (Supplementary Fig. 2b1), eventually leading to the shutdown of the AMOC. Meanwhile, aided by anomalous highs over the North Pacific and southern China (Supplementary Fig. 3b1), less atmospheric water vapor converges over the western tropical and subtropical Pacific (Supplementary Fig. 3a1), causing an increase in SSS there (Fig. 4i). The resulting high-salinity surface water is transported northward by the Kuroshio Current and further eastward by the Kuroshio Extension, thereby triggering the NPDW formation. The anomalous high over the North Pacific also leads to Ekman downwelling (Fig. 4e), which further aids the NPDW formation. These two processes collectively lead to the establishment of the PMOC in Flat.

In Flat, the thermohaline component of the PMOC in the South Indo-Pacific is weaker than that of the AMOC in Real (Fig. 3b2 vs a8). This can be attributed to the weaker Ekman pumping in the Southern Ocean in Flat in comparison to that in Real (Fig. 5a). The absence of the Antarctic topography induces a high-pressure anomaly over the Antarctic, which produces an anomalous northward pressure gradient, and thus an anomalous easterlies based on the geostrophic balance. This eventually results in an 80% reduction in wind stress and Ekman pumping along the Antarctic continent (Fig. 5a). The impact of this reduction on the MOC is evident in Fig. 3c1–c3, with the Deacon cell being weakened by 50% in Flat. As a result, the southward water mass transport in the intermediate-to-deep ocean across 30°S is about 30% weaker in Flat when compared to that in Real (Supplementary Fig. 4).

Once we recognize the changes in the MOC from Real to Flat, it becomes easy to understand the MOC changes in TP. From Flat to TP, the changes in the NH atmospheric circulation and moisture transport (Supplementary Fig. 3a4, b4) are similar to those from Flat to Real (Supplementary Fig. 3a2, b2). Consequently, we see similar changes in SSD, SSS, and Ekman pumping in TP as those in Real (Fig. 4b, f, j, and c, g, k). Specifically, the TP uplift leads to more net precipitation (Supplementary Fig. 2b3) and thus smaller SSD in the North Pacific, and a stronger Ekman upwelling in the North Pacific (Fig. 4g), which jointly shut down the PMOC. Moreover, the TP uplift causes less atmospheric moisture transport from the central tropical Pacific to the North Atlantic (Supplementary Fig. 3a4), resulting in accumulation of more saline surface water in the North Atlantic. However, the SSD in the North Atlantic (Fig. 4g) is still smaller than in Real (Fig. 4f) at this stage. The AMOC shows only a slight increase

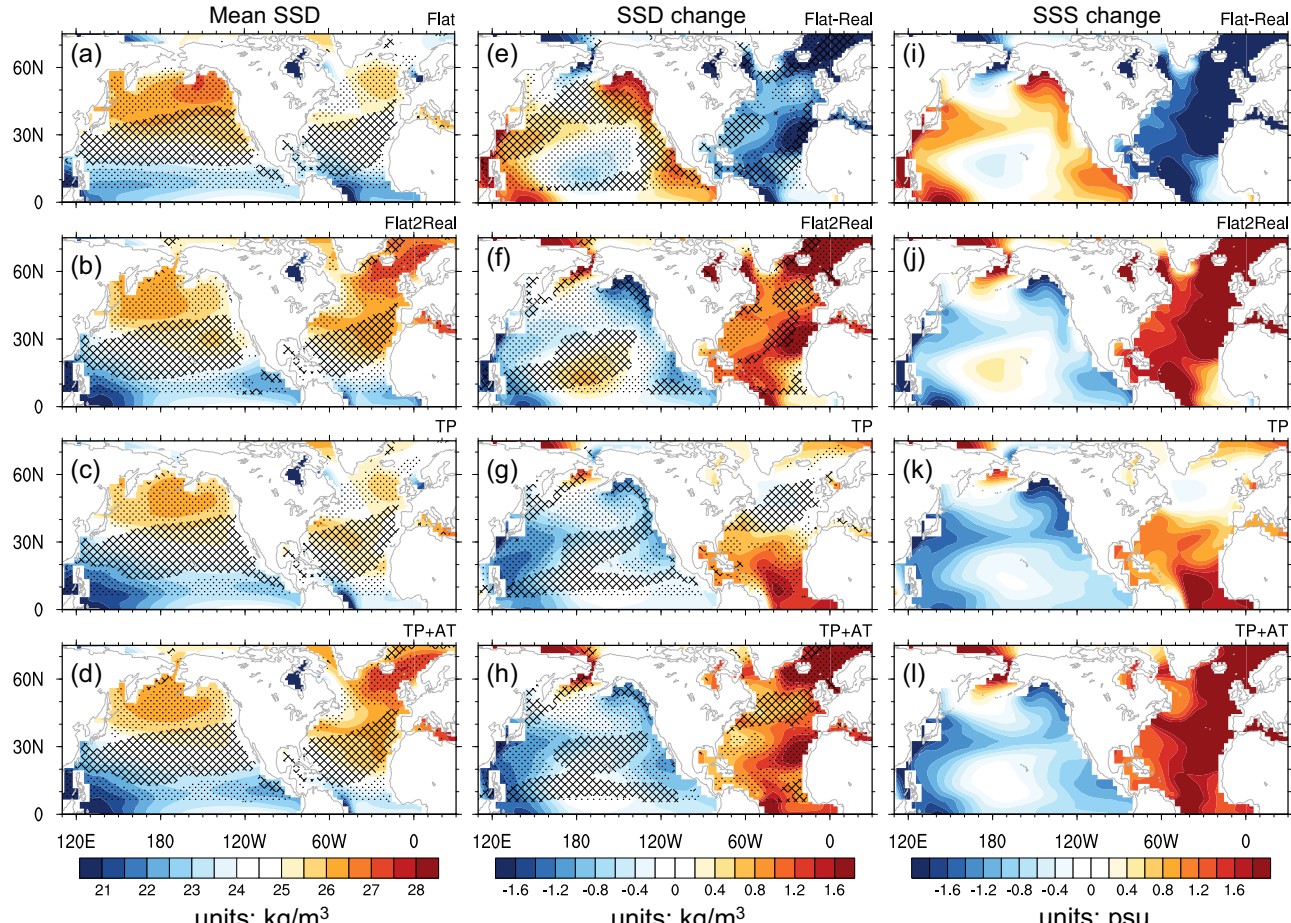

**Fig. 4 | Mean sea surface density (SSD), Ekman pumping, and their changes. a–d** Mean SSD (units: kg/m³) in Flat, Flat2Real, TP (Tibetan Plateau), and TP + AT (Antarctic), respectively, with black dots (crosses) denoting mean Ekman upwelling (downwelling) in each experiment. The Ekman pumping is calculated as $\omega_E = curl(\frac{\tau}{\rho f})$, where $\tau$ is the surface wind stress with units of $dyn/cm^2$, $\rho$ is the density of sea water ($1024 kg/m^3$), and $f$ is Coriolis parameter with units of $1/s$. **e** shows the equilibrium changes in SSD and Ekman pumping in Flat with respect to Real, while **f–h** show their equilibrium changes in Flat2Real (Year 5601–6000), TP, and TP + AT, respectively, with respect to Flat. Black dots (crosses) denoting region with anomalous Ekman upwelling (downwelling). **i** shows the equilibrium sea surface salinity (SSS) changes (units: psu) in Flat with respect to Real, while **j–l** show the equilibrium SSS changes in Flat2Real, TP, and TP + AT, respectively, with respect to Flat. Note that the patterns of **e** and **f** and those of **i** and **j** are nearly identical but with opposite signs. **e** and **i** are included for convenience of analysis. The mean SSD in experiments AT and AM is similar to that in Flat, and thus is not shown here.

(Fig. 2a). Furthermore, the Ekman pumping in the Southern Ocean in TP is much weaker than that in Real (Fig. 5b), which limits further development of the AMOC.

The presence of the AT alone does not alter the surface buoyancy in the NH (figure not shown). Similar to Flat, the NPDW formation occurs instead of the NADW formation. However, the Ekman pumping in the Southern Ocean in AT is as robust as that in Real (Fig. 5c), resulting in a more potent Deacon cell (Fig. 3f1) and a more forceful thermohaline component of the PMOC (Fig. 3e1) than those in Flat. Similar to Flat, the strong southward water mass transport occurs in the South Indo-Pacific, but not in the South Atlantic (Fig. 5g, Supplementary Fig. 4).

With the assistance of the AT, the establishment of the AMOC in TP + AT becomes possible (Fig. 3d8). The presence of the TP shuts down the PMOC, shifting the deep-water formation from the North Pacific to the North Atlantic. Simultaneously, the presence of the AT leads to strong Ekman pumping in the Southern Ocean, which enhances the NADW formation remotely (Fig. 4l). The MOC patterns in TP + AT are almost identical to those in Real (Fig. 3d8 vs a8) since both the surface buoyancy in the North Atlantic and the Ekman pumping in the Southern Ocean are almost identical to those in Real (Figs. 4d and 5d). Note that in TP + AT, there is strong southward

water mass transport in the South Atlantic, instead of in the South Indo-Pacific as in AT (Fig. 5g, h, Supplementary Fig. 4), because the presence of the TP alters the deep-water formation in the NH, which allows only the NADW to reach the Southern Ocean.

The presence of other large topographies, such as AM, RM, and GL, would not switch the MOC from the Pacific to the Atlantic. However, the PMOC in Exp AM is noticeably stronger than that in Flat (Fig. 2a). The presence of the AM reduces the equatorial trade wind, but amplifies the off-equatorial Ekman pumping (Supplementary Fig. 5a), thereby boosting the wind-driven STC in the South Indo-Pacific (Supplementary Fig. 5b) and augmenting the thermohaline component of the PMOC there, leading to a stronger PMOC (Fig. 3e2). On the other hand, both the RM and GL have minimal effects on the PMOC (Fig. 3a3–f3) because their presences do not significantly alter the global atmospheric moisture pattern[30] and the atmospheric circulation in the tropics (Supplementary Fig. 6).

The AMOC gradually increases in the first few hundred years after the TP uplift in both Flat2Real and TP + AT, followed by an acceleration and eventual return to a normal state in Real (Figs. 1 and 2b). The latter process is accompanied by a swift sea-ice loss in the subpolar Atlantic. In response to the TP uplift, more atmospheric moisture converges (diverges) over the North Pacific (Atlantic) (Supplementary Figs. 2b2,

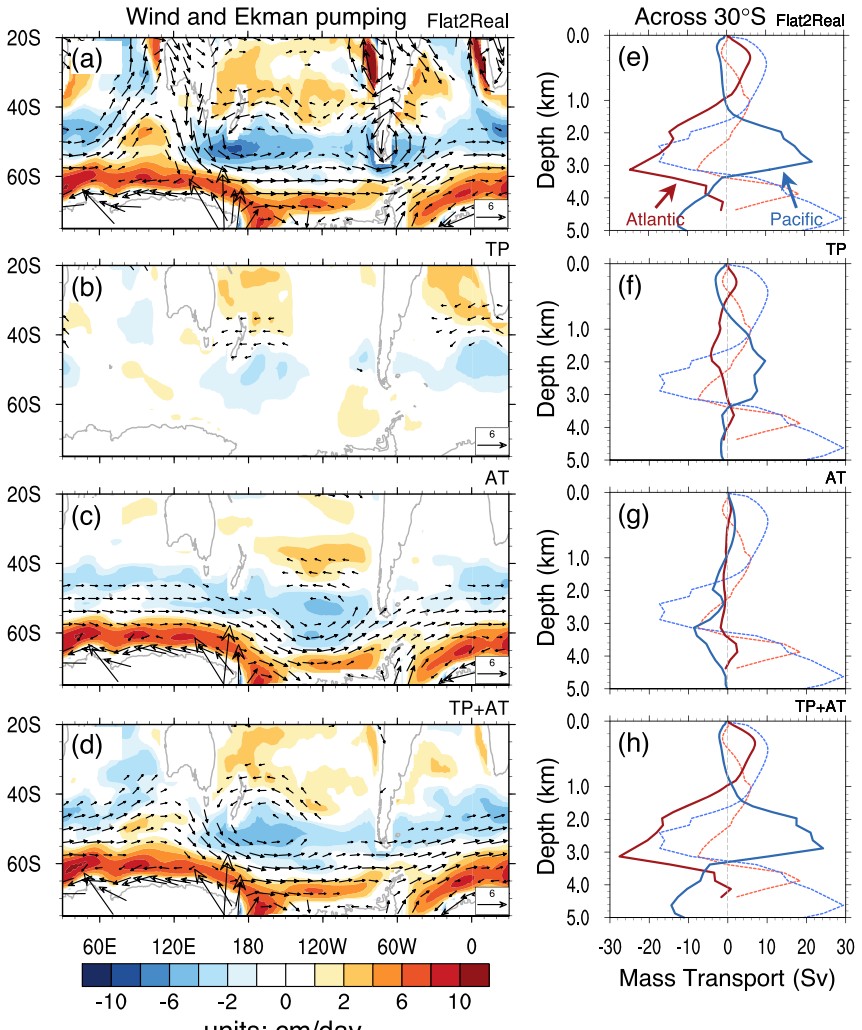

**Fig. 5 | Changes in wind, Ekman pumping, and mass transport. a–d** Equilibrium changes of wind stress at 850 hPa (vector; units: m/s) and Ekman pumping (shading; units: cm/day) in Flat2Real (Year 5601–6000), TP (Tibetan Plateau), AT (Antarctic), and TP + AT, respectively, with respect to Flat. Positive (negative) value indicates Ekman upwelling (downwelling). Ekman pumping is calculated using the surface wind stress as described in Fig. 4. **e–h** Equilibrium changes of zonally integrated net meridional water mass transport (units: Sv) across 30°S in Flat2Real, TP, AT, and TP + AT, respectively, with respect to Flat. The mean net meridional mass transport in Flat is plotted as a dotted curve, with the red and blue curves indicating the Atlantic and Pacific, respectively. Positive value represents northward transport. The vertical coordinate in **e–h** is depth (units: km).

b5 and 3a2, a3), shutting down the PMOC and triggering a gradual increase of the AMOC at the same time. The latter enhances the northward heat transport in the Atlantic, leading to a gradual retreat of sea ice in the subpolar Atlantic. The sea ice retreat is also helped by the anomalous northward Ekman flow, forced by the anomalous easterlies over the subpolar Atlantic (Supplementary Fig. 3b2).

The evolution of sea ice in Flat2Real is shown in Fig. 6. The northward retreat of sea ice is illustrated by the sea ice velocity in Fig. 6b, which leads to additional freshwater loss in this region (Supplementary Fig. 2b2, b5). This freshwater loss in turn increases SSD, the NADW formation and thus the AMOC consequently. In the first few hundred years after adding the TP, the sea-ice in the subpolar Atlantic retreats northward slightly (Fig. 6b), and the March mixed layer depth (MLD) deepens slightly (Fig. 6a), corresponding to a gradual increase of the AMOC. About 500 years after the TP uplift, the collective effects of accumulated saline water in the subpolar Atlantic and Ekman pumping in Southern Ocean accelerate the AMOC, so that the sea-ice margin retreats rapidly northward (Fig. 6b, dashed red curve), resulting in a large amount of freshwater flux loss in the subpolar Atlantic, a rapid deepening of the MLD (Fig. 6a), and a further enhancement of

the AMOC (Figs. 1 and 2b). The sea-ice margin reaches its quasi-equilibrium roughly 1000 years after the TP uplift (Fig. 6b), accompanied by significant sea-ice melting in the GIN seas. The sea-ice evolution in TP + AT (Supplementary Fig. 7a, b) displays a similar pattern to that in Flat2Real, while changes in sea ice and MLD in TP are minimal (Supplementary Fig. 7c, d). The evolutions of AMOC, the MLD and sea-ice coverage and margin in Flat2Real and TP + AT suggest a positive feedback between the AMOC and sea ice changes, which eventually leads to the establishment of AMOC. This feedback has been shown in many previous studies[34,47].

## Discussion
Continental topography plays a vital role in shaping Earth's climate. By conducting topography perturbation experiments, we can observe the dynamic processes by which atmospheric freshwater converges over the Pacific and diverges over the Atlantic in response to the TP uplift. These processes shut down the PMOC and enhance the NADW formation. The latter process then starts a positive feedback loop between the AMOC and sea ice in the subpolar Atlantic, pushing the final establishment of the AMOC. The TP acts as a giant attractor of

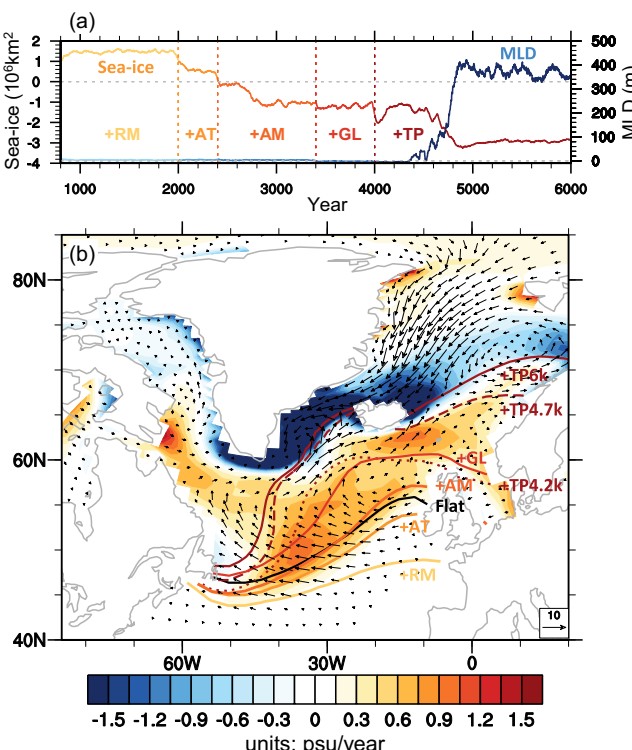

**Fig. 6 | Changes in sea ice and mixed layer depth (MLD) with the sequential uplift of different mountains in Flat2Real. a** Temporal evolution of the changes in sea-ice cover in the Arctic (units: $10^6$ km², left ordinate) and MLD (units: m, right ordinate) in the subpolar Atlantic, with respect to Flat. The sea-ice cover is annual averaged. The MLD is for March and is calculated using the method of Large et al.[57], which represents the site of the deepest vertical mixing and convection, and thus deep-water formation. **b** Sea-ice margin (curve) at different stages of Flat2Real, and equilibrium change in virtual salt flux (VSF) due to sea-ice formation or melting (units: psu/year; shading), and sea-ice velocity (units: cm/s; vector), with respect to Flat. These changes are annual averaged. The sea-ice margin is defined by the 15% sea-ice fraction; and different colors show sea-ice margins at different stages of the uplift. Changes in sea-ice velocity and VSF are obtained by subtracting Flat from the last 200-year-averaged values of Flat2Real. Positive (negative) VSF indicates loss (gain) of fresh water in the ocean.

fresh water in the NH, while the Antarctic continent acts as a giant draught-fan engine that forces Ekman pumping in the Southern Ocean, contributing to the establishment of the global ocean conveyor belt. Although no single mountain range can lead to the full establishment of the AMOC, the TP alone can shut down the PMOC. The presence of the TP can modulate the global hydrological cycle in a fundamental way, which may have shaped the modern-day thermohaline circulation.

Our experiments illustrate the dynamic contribution of Ekman pumping in the Southern Ocean to the GMOC, which is consistent with previous findings[43,44,48]. However, this Ekman pumping can pump water from either the North Pacific or the North Atlantic, depending on in which basin the NH deep-water formation occurs. Only after the TP uplift, the change in global hydrological cycle leads to more saline water accumulated in the North Atlantic, initializing the NADW formation. Assisted by strong Ekman pumping due to the AT, the AMOC can be established. In addition, our topography experiments show a clear planetary wave train in the NH mid-to-high latitudes (Supplementary Fig. 3) connecting the Eurasian continent, the North Pacific, the North American continent, and the North Atlantic as the atmospheric circulation changes with the addition of the TP. This wave train structure is similar to that observed from Flat to Real, highlighting TP's

global effects. No other topography can force this kind of wave train structure in the NH.

The results presented here are a first step toward a better understanding of the coupled impacts of the presence or absence of various major mountain ranges on the GMOC, as well as their cumulative impact under modern conditions. Our experiments are specifically designed to disentangle the impact of topography on ocean circulation from other contributing factors, thereby keeping the bathymetry and continental configuration, as well as greenhouse gases, incident solar radiation, and orbital parameters to their modern state in our experimental setup. The individual mountain uplift experiments in this study allow us to explore the links and teleconnections between uplift regions and the climate of the modern world. Moreover, the sequential mountain uplift experiment Flat2Real may offer interesting insights about the long-term impact of mountains on Cenozoic climate change, notwithstanding significant caveats.

Previous studies suggested that the TP uplift played a key role in shaping Cenozoic climate through circulation changes and weathering[13,49,50]: over the past 40 million years, this uplift led to substantial deflection of the atmospheric jet stream, intensified monsoonal circulation, increased rainfall on the front slopes of the Himalayas, and conductive conditions for the formation of deep and intermediate waters in the North Atlantic. By using experiment Flat as a starting point and tracing the changes due to various uplifts, we can quantify the linkages between uplifts and climate changes and understand how the MOC changes as the progressive uplift of continental mountains.

It should be noted that the conclusions of this study may have some limitations due to the model employed. For instance, the model resolution may not be optimal for analyzing the impact of topography on the global climate, as models with coarse resolution may not adequately reproduce the effects of mountain ranges with complex topography, particularly those with an elongated shape such as the Andes or the Rockies. Additionally, we did not consider the effects of continental drift and oceanic gateway switches, nor did we treat Greenland and Antarctic glaciation dynamically. The setup of modern land-sea mask and ocean gateways in our experiments implies that the TP impact on the GMOC could be exaggerated. The Tethys Seaway closing, the Panama Isthmus closing and the Bering Strait opening all occurred after the TP uplift[8], all of which may lead to significant changes in ocean circulation and modulate NADW formation[51–53]. Furthermore, the background climate in the experiments uses the preindustrial conditions with constant $CO_2$, and the effect of chemical erosion in rapidly uplifted areas on atmospheric $CO_2$ is not accounted for. Typically, atmospheric $CO_2$ levels cannot remain in a steady state during the time of intense tectonism, which can last for millions of years, due to the temperature-weathering feedback mechanism. The TP uplift may have resulted in a decrease of atmospheric $CO_2$ level, which can also enhance the NADW formation due to the growth of large continental ice sheets in the NH[49]. To better understand the uplift effect on the global climate, it is essential to gain a quantitative understanding of the long-term carbon cycle. Overall, there are many factors at play in long-term climate change, and their effects remain poorly constrained.

We still want to emphasize that our model results capture the basic direction of climate change. The uplift of giant mountains undoubtedly played a significant role in the evolution of global thermohaline circulation; and our results align with those of other orography experiments conducted using different models[14–16,40]. However, the impact of giant mountains on the AMOC remains a topic of debate, and our results contribute to a better understanding of the controls exerted by mountain ranges over the MOC, which is critical for a better comprehension of the role of the ocean in past and future climate transitions.

## Methods

We use the National Center for Atmospheric Research Community Earth System Model version 1.0 (NCAR CESM 1.0) in this study. The CESM1.0 employed here has the grid of T31_gx3v7, which consists of the atmospheric component (CAM4) with 26 vertical levels and T31 horizontal resolution (3.75° × 3.75°); the ocean component (POP2) with 60 vertical levels and gx3v7 horizontal resolution (approximate 3° near the poles to 0.6° at the equator); the land component (CLM4) and the sea-ice component (CICE) with the same horizontal resolution as the CAM4 and POP2, respectively. More details about these model components can be found in Gent et al.[54] and Shields et al.[55].

To compare the impacts of various mountain ranges on the global MOC, we conduct a few sets of topography experiments (Table 1, Supplementary Fig. 1). These experiments are designed in relation to the "Flat" experiment, which features a globally flat topography at 50 m above the sea level (Supplementary Fig. 1a). Although not precisely representative of any specific Earth epoch, Flat serves as a reference point for the other experiments. Flat is integrated for 1600 years under the preindustrial conditions, starting from a control run with realistic modern topography (Real) that was completed previously[56]. We examine the effects of five prominent topographies in this study: the Rocky Mountains, the Antarctic continent, the Andes Mountains, Greenland, and the Tibetan Plateau. Except for topography height, all other boundary conditions remain the same as in Flat or Real. The ocean-land configuration is set to modern-day conditions without correction for plate tectonic motion. Atmospheric $CO_2$ concentration is maintained at the preindustrial level (285 ppm). Changes in river routing and vegetation type are not considered. Continental ice sheets are treated as bright rocks in the model. The planetary albedo can adjust itself according to thermal conditions. These experiments are conducted as single-variable (orography) sensitivity tests, rather than paleoclimate simulations involving multiple prescribed geologic boundary conditions.

The first set of experiments includes adding each of the five different topography individually to the Flat. These experiments are named RM (Rocky Mountains), AT (Antarctic), AM (Andes Mountains), GL (Greenland), and TP (Tibetan Plateau), respectively. Each topography is added to Flat starting from year 801; and each experiment is then integrated for at least 800 years. The second set of experiments, named Flat2Real, includes adding the five topography sequentially according to their uplift times. The RM is added to Flat in year 801; and the model is integrated for 1200 years. The AT is added after the RM in year 2001; and the model is integrated for 400 years. The AM is added in year 2401; and the model is integrated for 1000 years. The GL is added in year 3401; and the model is integrated for 600 years. The TP is added in year 4001; and the model is integrated for 2000 years. The third set of experiments combines topography differently, such as TP + RM, TP + AT, TP + AM, and TP + GL, plus Real with all five mountains. All these experiments start from year 801 of Flat, and are integrated for at least 800 years. More detailed information can be found in Table 1.

The changes due to the presence of unique topography are obtained by subtracting the results of Flat from each topography experiment. Some experiments show an initial jump when adding the topography suddenly, but that jump has little effect on the equilibrium state. These experiments indicate that uplift causes a wide array of changes in the global climate. We use student-$t$ test to examine the statistical significance of our results. Most changes are significant at the 95% confidence level, which is expected because altering large topography induces strong mechanical forcing and obvious responses around the globe. For visual clarity, significance test is not included in any figures.

## Data availability

Model output in this study is available from the corresponding author upon request.

## Code availability

Code used in this study is available from the corresponding author upon request.

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

## Acknowledgements

This work is supported by the NSF of China (Nos. 91737204, 91937302, 42288101, and 42230403) and by the foundation at Shanghai Scientific Frontier Base for Ocean-Atmosphere Interaction Studies. The experiments were performed on the supercomputers at the National Supercomputer Centre in Tianjin (Tian-He No.1).

## Author contributions

H. Yang designed the experiments, directed all analyses and wrote the paper; R. Jiang performed the experiments, analyzed the results and plotted all figures. Q. Wen, Y. Liu, G. Wu and J. Huang presented advices in the experiments' design and helped to revise the manuscript.

## Competing interests

The authors declare no competing interests.
