## [Peer Review File · Nature Communications]

The Role of Mountains in Shaping the Global Meridional Overturning CirculationEditorial Note: Parts of this Peer Review File have been redacted as indicated to remove third-party material where no permission to publish could be obtained.

REVIEWER COMMENTS

Reviewer #1 (Remarks to the Author):

This is a review of “Which Mountains Matter to Global Meridional Overturning Circulation” by Haijun Yang and coauthors for potential publication in Nature Communications.

Using numerical simulations with a low-resolution Earth System Model (CESM1.0 at $\sim 3^\circ$), the authors investigate how the absence/presence of major mountain ranges (the Tibetan Plateau/Himalaya, the Antarctic topography, the Andes, the Rockies and the Greenland topography) alters the global meridional overturning circulation. They show that in a “flat” Earth, the AMOC shuts down in favour of the PMOC, and that adding new reliefs to this flat Earth may spectacularly reorganize the global MOC. In particular they show that the presence of the Tibetan Plateau is both a sufficient condition to suppress the PMOC and a necessary (but not sufficient) condition to the existence of the AMOC. They show that the Andes and Antarctic Mountains are reinforcing drivers of, respectively, the PMOC (but not the AMOC) and both the AMOC or PMOC (depending on the presence or not of the Tibetan Plateau). On the contrary, the Rockies and the Greenland topography only exhibit weak impacts on the MOC in the simulations presented here. Finally, an interesting aspect is that the authors perform what can be seen as a more realistic scenario of the Cenozoic mountain evolution (Flat2Real), in which they sequentially add the aforementioned mountain ranges to a flat Earth and tentatively suggest that the main phases of uplift during the Cenozoic may have at least partly driven the Cenozoic MOC evolution, though it must be noted that the authors state the limitations of their approach to the “real” history of the MOC.

I must say I really enjoyed reading this manuscript. The experiments are nicely designed and the results are robustly demonstrated. The manuscript is also clear and well-written. I would quite straightforwardly recommend this manuscript for publication from a physics (mechanistic) perspective. However, and in particular in the context of a broadly oriented journal like Nature Communications, I feel like the rationale behind the experiments and the contribution of the findings to the understanding of long-term Cenozoic evolution should be better discussed and/or articulated (see below) and I therefore recommend to return the manuscript to the authors with minor revisions. I hope this can help strengthen the manuscript for broader impact and I would be happy to read a revised version.

Two points must be better treated.

First, while I think the authors nicely show how the Tibetan Plateau and Antarctic Mountains make the system switch from the PMOC to the AMOC, the chronology of MOC changes and uplift events described in l. 7-14 and 24-33 overlooks the (very) large uncertainties associated with these. For instance, some studies argue that parts of the Tibetan Plateau were already high in the late Eocene (e.g., Xiong et al. 2020, Su et al. 2019). The Transantarctic and Gamburtsev Mountains over Antarctica were likely already present at the start of the Cenozoic (Elliot 2014, Ferraccioli 2011)—except if the Antarctic uplift considered here is in fact the Eocene-Oligocene glaciation of Antarctica? In this case, it should be more explicit. The history of the AMOC/PMOC is also quite debated and not really agreed upon. l. 7-14 and 24-33 instead convey another impression upon reading. I agree that it is not at all the job of this manuscript to provide a comprehensive review of these changes but these sections should be re-written with more

caveats to reflect these uncertainties.

Second, though this is touched upon on l. 18-19 and 235-236, the impacts of oceanic gateways cannot simply be neglected because these have the potential to counterbalance the effects of mountain uplift by providing pathways to efficiently mix water masses. I think that mentioning some relevant work relative to the Cenozoic evolution of a few major gateways—e.g., Drake Passage/Tasman Seaway, Panama, Arctic gateways—and how they may have modulated the ocean circulation would be worthwhile. For instance, before the deep opening of Drake Passage / Tasman Seaway in the late Eocene (e.g., Sauermilch et al. 2021), it is likely that the Ekman pumping due to the Westerlies may have been much weaker. This is especially relevant as some simulation findings contradict the history of MOC changes as written l. 7-14 and could be used to provide alternative hypothesis: still using the Drake/Tasman example, Fig. 2 shows that adding the Antarctic Mountains substantially increase the PMOC intensity (by the way this is not seen on Flat2Real because the simulation is only run for 400 years after the AT addition and before the AM addition, which is not enough in particular because the AM drives a similar increase in PMOC intensity and obscures the AT effect). However, chronologically, the PMOC is expected to decrease during the Cenozoic (l. 10 of the manuscript, see also Ferreira et al. 2018) although this is quite uncertain. In this case, a counterbalancing effect of the Southern Ocean gateways could provide a hypothesis to limit the impact of AT on the MOC.

Now I am well aware of how delicate it is to speculate about such changes and I am not asking the authors to overly expand on these kinds of hypothesis but I think the authors should still use a couple of judiciously-chosen example to make clear that repeating these experiments with another land-sea mask (e.g. narrow Drake/Tasman and wide Panama gateways) could give a completely different story, including a possibly much weaker role for the Tibetan Plateau.

Minor points.

l. 113 Why is the salt flux virtual?

Throughout. How is computed Ekman pumping/downwelling?

l. 165-166. Not only surface buoyancy in North Atlantic but also Ekman pumping in the Southern Ocean.

l. 176-177. This sentence reads a bit weirdly because the AM does not alter the moisture pattern either but still affect the MOC intensity via alteration of the atmosphere dynamics.

Methods. Are changes in runoff consequent in the simulations? What would happen if the runoff was adequately re-routed?

Fig. 4. I get what dots and crosses represent on Fig. 4a1-a4 but not on Fig. 4b1-b4 (the difference figures). Do crosses (dots) represent increased downwelling (upwelling)?

Also what do you mean by “mean Ekman pumping/downwelling”? I do not understand how dots and crosses can have units of cm/day.

Fig. 5. Mass transport on Fig. 5b1-b4 should be in Sv to be consistent with the other figures.

Fig. 6. What month or season is reflected in the sea-ice margin/velocity? It should be clearly written in the legend.

Extended data Fig. 1 should be plotted at model resolution because this is what matters in the results.

References.

Elliot, D. H. (2013). The geological and tectonic evolution of the Transantarctic Mountains: a review. Geological Society, London, Special Publications, 381(1), 7-35.

Ferraccioli, F., et al. (2011). East Antarctic rifting triggers uplift of the Gamburtsev Mountains. *Nature*, 479(7373), 388-392.

Ferreira, D., et al. (2018). Atlantic-Pacific asymmetry in deep water formation. *Annual Review of Earth and Planetary Sciences*, 46, 327-352.

Sauermilch, I., et al. (2021). Gateway-driven weakening of ocean gyres leads to Southern Ocean cooling. *Nature communications*, 12(1), 6465.

Su, T., et al. (2019). Uplift, climate and biotic changes at the Eocene–Oligocene transition in south-eastern Tibet. *National Science Review*, 6(3), 495-504.

Xiong, Z., et al. (2020). The early Eocene rise of the Gonjo Basin, SE Tibet: From low desert to high forest. *Earth and Planetary Science Letters*, 543, 116312.

Reviewer #2 (Remarks to the Author):

The manuscript by Haijun Yang et al. presents an investigation into the influence of orography on the Meridional Overturning Circulation (MOC) using the low-resolution coupled model CESM. Previous studies, including Fallah et al. (2016), Su et al. (2018), and Yang and Wen (2020), have examined the effects of the Tibetan plateau on oceanic circulation. Maffre et al. (2017) conducted an analysis on the impact of global orography on oceanic circulation by comparing two simulations: one incorporating present-day geography and another using a flat Earth representation. In this regard, Haijun Yang et al.'s work combines these two approaches by investigating the effects of a flat Earth representation and various combinations of present-day orography on oceanic circulation. While this study is interesting, I have a few concerns regarding this approach.

(L24-L39) One concern with the sensitivity experiments conducted in this study is that they include the

entire topography of the continent, rather than focusing solely on specific features like the Tibetan plateau or the Rocky Mountains. Previous research has demonstrated that the uplift of the Tibetan plateau influences the Atlantic Meridional Overturning Circulation (AMOC). However, by incorporating the entire topography of a continent, it becomes challenging to isolate the effects of individual landforms and determine whether minor features play a role in the establishment or disruption of the AMOC/PMOC.

Additionally, the study includes the Andes Mountains and the entire South American, African, Middle Eastern regions up to the Zagros Mountains and Anatolian plateau under the scenario "AM". Regarding the ice sheet (Antarctica and Greenland), the study tests the sensitivity to changes in elevation, but it is not specified whether the authors maintained the albedo of ice when removing the Antarctica / Greenland topography (the term ice sheet is never employed in the manuscript). This information is crucial as altering the albedo would affect the energy balance and potentially influence the circulation patterns and precipitation.

Moreover, when orography is removed, the drainage basin system is expected to undergo changes. The authors indicate that river routing is kept unchanged. Understanding the modifications in the drainage basin system is relevant to comprehending the overall impact of orography removal on oceanic circulation.

These concerns regarding the specificity of landforms, albedo adjustments, and drainage basin modifications should be addressed in the manuscript to provide a comprehensive analysis of the study's findings.

Despite the fact that the authors exclude to explain the evolution of AMOC during the Cenozoic and restrain this explanation to the role of Tibetan plateau in the present day geography, the Figure 1 exhibits the impact of a sequential uplift of mountains with a "geological" chronology. The sequence is supposed to mimic the main phase of uplift (or growth of ice sheet).

Although the authors do not try to provide a detailed explanation of the evolution of the AMOC/PMOC during the Cenozoic era and instead focus on the role of the topography in present-day geography, Figure 1 demonstrates the impact of sequential mountain uplift with a geological chronology. The sequence is intended to simulate the impact on ocean circulation of the primary phase of uplift or the growth of an ice sheet.

The length of the simulation is a matter of debate. Following the addition of Antarctic, the simulation is integrated for 400 years. However, this duration may be considered relatively short to ensure that the deep ocean layers have fully reached equilibrium with the boundary conditions.

(L69) The authors do not provide an explanation regarding whether the albedo was modified when removing the topography of Antarctica. It would be beneficial to clarify whether any adjustments were made to account for changes in albedo resulting from the modification of Antarctic topography.

(L93-...) The authors primarily focus on describing the oceanic changes in their study, driven by atmospheric modifications resulting from the addition or removal of topography. However, they do not provide a detailed explanation of the underlying mechanisms involved. It would indeed be valuable to include a few sentences elucidating the specific mechanisms by which the evaporation or precipitation patterns are altered when modifying the topography.

In this work, the authors do not explicitly mention conducting statistical significance tests to evaluate the differences between two variables.

(L112-122) The mechanism causing a change in precipitation or evaporation (or both) are discussed. The changes in atmospheric circulation and moisture transport are associated to changes in GPH. The authors do not explicitly delve into the topic of temperature changes in relation to GPH variations, nor do they discuss the potential role of Sea Surface Temperature (SST) changes in the North Atlantic affecting the North Atlantic Deep Water (NADW) circulation. To provide a more comprehensive analysis, it would be beneficial for the authors to address the possible temperature changes resulting from GPH modifications and discuss the influence of SST changes on NADW formation explicitly. Incorporating these aspects would further elucidate the connections between atmospheric circulation, temperature, and oceanic processes, enhancing the overall understanding of the research findings.

(L122-L123) “The expansion of sea ice (...) carries a substantial amount of freshwater”: the sentence is not clear to me.

(L135) “The absence of the Antarctic topography leads to an 80% decrease in wind stress”: is it due to temperature gradient?

Figure 5: the Ekman pumping/downwelling is discussed with the 850hPa wind stress. This is not clear. Why don't the authors show the wind stress at the sea surface ?

Positive mass transport across 30°S is counted positively northwards. This is not mentioned in the figure caption.

(L202+Extended Data Figure 3a1-4) The authors' conclusion that the Tibetan plateau acts as a significant attractor of freshwater in the Northern Hemisphere (NH) is potentially accurate if the boundary conditions used in their scenario are limited to the Tibetan plateau alone. However, it is important to consider that changes in the North Pacific region can also be influenced by the westerlies blowing from Eastern Asia, which may be impacted by the presence or absence of landforms located to the north of the Tibetan plateau.

(L224) I think that the authors could quote a more recent paper than the Ruddiman's 1989 paper
(L232-243) I agree with the observation regarding the limitations of the study, particularly with respect to the spatial resolution of the atmospheric model. I wonder whether the low resolution of the model might not be a potential factor contributing to the lack of impact observed for certain mountain ranges, particularly those with an elongated shape such as the Andes or the Rockies. Mountain ranges with complex topography may not be adequately captured or represented in the model due to the coarse resolution. In contrast, the Antarctic ice cap and the Tibetan plateau may be better resolved within the model. This could explain why their impacts are more apparent in the simulations.

Other comment

(L24-33) The timing of uplift is also debatable. A part of the Tibetan plateau is largely uplifted before the Late Miocene.

Antarctic (ice sheet ?) : the authors explains that “Antarctic rapidly expanded in the Oligocene and persisted until the Late Oligocene”. This is not clear to me what the authors mean by “persisted until the Late Oligocene”. Does it mean that the ice sheet melts after the Late Oligocene ? No reference is provided.

Andes uplift => reference 19 : Zachos et al. 2001 ?

Figure 5: continent contours are too thin to be seen when printed

Extended Data Figure 3 : continent contours are hard to see

Replies to Reviewer #1:

Thank you very much for these constructive comments. We have revised the manuscript carefully based on these suggestions. The following are our point-by-point replies.

Using numerical simulations with a low-resolution Earth System Model (CESM1.0 at $\sim 3^\circ$), the authors investigate how the absence/presence of major mountain ranges (the Tibetan Plateau/Himalaya, the Antarctic topography, the Andes, the Rockies and the Greenland topography) alters the global meridional overturning circulation. They show that in a “flat” Earth, the AMOC shuts down in favour of the PMOC, and that adding new reliefs to this flat Earth may spectacularly reorganize the global MOC. In particular they show that the presence of the Tibetan Plateau is both a sufficient condition to suppress the PMOC and a necessary (but not sufficient) condition to the existence of the AMOC. They show that the Andes and Antarctic Mountains are reinforcing drivers of, respectively, the PMOC (but not the AMOC) and both the AMOC or PMOC (depending on the presence or not of the Tibetan Plateau). On the contrary, the Rockies and the Greenland topography only exhibit weak impacts on the MOC in the simulations presented here. Finally, an interesting aspect is that the authors perform what can be seen as a more realistic scenario of the Cenozoic mountain evolution (Flat2Real), in which they sequentially add the aforementioned mountain ranges to a flat Earth and tentatively suggest that the main phases of uplift during the Cenozoic (Since 65 Ma) may have at least partly driven the Cenozoic MOC evolution, though it must be noted that the authors state the limitations of their approach to the “real” history of the MOC.

I must say I really enjoyed reading this manuscript. The experiments are nicely designed and the results are robustly demonstrated. The manuscript is also clear and well-written. I would quite straightforwardly recommend this manuscript for publication from a physics (mechanistic) perspective. However, and in particular in the context of a broadly oriented journal like Nature Communications, I feel like the rationale behind the experiments and the contribution of the findings to the understanding of long-term Cenozoic evolution should be better discussed and/or articulated (see below) and I therefore recommend to return the manuscript to the authors with minor revisions. I hope this can help strengthen the manuscript for broader impact and I would be happy to read a revised version.

Responses: Thank you very much for your valuable comments, which help us improve the manuscript tremendously. Considering the comments from all the reviewers, we revised the manuscript primarily in these following aspects:

- 1) Introduction to research background is much improved. We rewrote statements and make it clear to readers that there are many controversies regarding the on/off state of the AMOC/PMOC, as well as the timing of mountain uplift.
- 2) Mechanisms for global climate changes in response to mountain uplift are explained with more details;
- 3) Some relevant references are added; and most of figures are replotted.

Specific comments:

1. *First, while I think the authors nicely show how the Tibetan Plateau and Antarctic Mountains make the system switch from the PMOC to the AMOC, the chronology of MOC changes and uplift events described in l. 7-14 and 24-33 overlooks the (very) large uncertainties associated with these. For instance, some studies argue that parts of the Tibetan Plateau were already high in the late Eocene (37.9-33.7 Ma) (e.g., Xiong et al. 2020, Su et al. 2019). The Transantarctic and Gamburtsev Mountains over Antarctica were likely already present at the start of the Cenozoic (65 Ma) (Elliot 2014, Ferraccioli 2011), except if the Antarctic uplift considered here is in fact the Eocene-Oligocene (56-23 Ma) glaciation of Antarctica? In this case, it should be more explicit. The history of the AMOC/PMOC is also quite debated and not really agreed upon. l. 7-14 and 24-33 instead convey another impression upon reading. I agree that it is not at all the job of this manuscript to provide a comprehensive review of these changes but these sections should be re-written with more caveats to reflect these uncertainties.*

Responses: Thank you very much for raising these important questions. We rewrote this part and made it clear to readers that there are many controversies on the on/off state of the AMOC/PMOC in the history in Lines 7-15: “*Geological evidence reveals that the primary deep-water formation region in the Northern Hemisphere (NH) might have undergone a shift from the Pacific to the Atlantic in the past. Some studies suggest that North Pacific deep-water (NPDW) formation was strong during the Paleocene period (about 65-55 Ma, million years ago), while North Atlantic deep-water (NADW) formation was weak and likely begun to develop during the early Oligocene period (about 35-33 Ma). Consequently, the modern AMOC might initially develop in the late Miocene (about 12-9 Ma) and be fully established until the late Pliocene to early Pleistocene period (about 4-3 Ma). Nonetheless, the actual evolutionary history of the AMOC and PMOC remains a topic of considerable debate.*”

As far as the uplift timings of different mountains are concerned, we rewrote the statement in Lines 27-40 as follows: “... *Although the transantarctic and Gamburtsev Mountains over Antarctica were likely already present at the start of the Cenozoic (65 Ma) (Elliot 2014, Ferraccioli 2011), the glaciation of Antarctica was thought to occur during the Eocene-Oligocene (56-23 Ma), predating the uplift of the Andes Mountains (AM) that rose around 24 Ma...*” “... *Although some studies argue that parts of the TP were already formed in the late Eocene (38-33 Ma) (e.g., Xiong et al. 2020, Su et al. 2019), the rapid and main uplift of TP was established about 10 Ma. This timing coincided with the onset of NADW formation, suggesting a potential connection between these two mountains uplifts and the development of NADW. Still, it’s important to note that the timing of the uplift of major global mountain ranges remains a highly debated topic.*”

Please refer to Lines 27-40 in the revised manuscript. Due to the article length limited by the journal, we apology that we are unable to include detailed description on the uncertainties in the manuscript.

- 2. Second, though this is touched upon on l. 18-19 and 235-236, the impacts of oceanic gateways cannot simply be neglected because these have the potential to counterbalance the effects of mountain uplift by providing pathways to efficiently mix water masses. I think that mentioning some relevant work relative to the Cenozoic evolution of a few major gateways, e.g., Drake Passage/Tasman Seaway, Panama, Arctic gateways, and how they may have modulated the ocean circulation would be worthwhile. For instance, before the deep opening of Drake Passage / Tasman Seaway in the late Eocene (e.g., Sauermilch et al. 2021), it is likely that the Ekman pumping due to the Westerlies may have been much weaker. This is especially relevant as some simulation findings contradict the history of MOC changes as written l. 7-14 and could be used to provide alternative hypothesis: still using the Drake/Tasman example, Fig. 2 shows that adding the Antarctic Mountains substantially increase the PMOC intensity (by the way this is not seen on Flat2Real because the simulation is only run for 400 years after the AT addition and before the AM addition, which is not enough in particular because the AM drives a similar increase in PMOC intensity and obscures the AT effect). However, chronologically, the PMOC is expected to decrease during the Cenozoic (l. 10 of the manuscript, see also Ferreira et al. 2018) although this is quite uncertain. In this case, a counterbalancing effect of the Southern Ocean gateways could provide a hypothesis to limit the impact of AT on the MOC.*

Responses: Thank you very much for your insightful comments. You proposed a very interesting question that deserves an in-depth investigation. Following your comments, we are going to design experiments to study individual contributions of open/closed Drake Passage/Tasman seaway and the AT/AM topography height, and their combined effect on the global MOCs.

Since the manuscript is limited by words, we only briefly mention that “...*ocean basin geometry, ocean gateways and continental topography may contribute to different overturning modes. ... The opening of the Drake Passage/Tasman seaway in the late Eocene may have promoted the NADW formation and AMOC formation...*” in lines 19-23.

3. *Now I am well aware of how delicate it is to speculate about such changes and I am not asking the authors to overly expand on these kinds of hypothesis but I think the authors should still use a couple of judiciously-chosen example to make clear that repeating these experiments with another land-sea mask (e.g. narrow Drake/Tasman and wide Panama gateways) could give a completely different story, including a possibly much weaker role for the Tibetan Plateau.*

Responses: Thank you very much for these insightful comments. This work is just a beginning; and we are actually starting to do lots of other experiments to enhance our understanding of the topography, the ocean and land gateway's roles in the ocean circulation. There are so much to do. We would like to show you our future work, if possible, based on your suggestion of the experiments.

The purpose of this work is to assess the roles of different mountains in the GMOC under present-day situation. We try to focus on comparing the roles of different mountains, not on comparing mountains' roles with the roles of other factors (such like ocean gateway, land gateway, etc.). We totally agree with you that there would be many possibilities and the outcomes could be completely different if we try to mimic “real” evolution of paleoclimate.

Minor points.

1. *1. 113 Why is the salt flux virtual?*

Responses: It actually is “freshwater flux” across the ocean surface, not really salt flux; so, it is called “virtual salt flux.”

2. *Throughout. How is computed Ekman pumping/downwelling?*

Responses: The Ekman pumping is calculated using the wind stress at the ocean surface. $\omega_E = \text{curl}\left(\frac{\tau}{\rho f}\right)$, where τ is the surface wind stress with the units of N/m^2 , ρ is the density of sea water ($\rho = 1024 \text{ kg}/\text{m}^3$), and f is Coriolis parameter with the units of $1/\text{s}$. So, the units of

$$\omega_E \sim \frac{1}{\text{m}} \left(\frac{\frac{\text{N}}{\text{m}^2}}{\frac{\text{kg}}{\text{m}^3} \cdot \text{s}^{-1}} \right) \sim \frac{1}{\text{m}} \left(\frac{\frac{\text{kgm}/\text{s}^2}{\text{m}^2}}{\frac{\text{kg}}{\text{m}^3} \cdot \text{s}^{-1}} \right) = \text{m}/\text{s} \rightarrow \text{cm}/\text{day}$$

In the caption of Fig. 4, we added how the Ekman pumping is calculated.

3. *l. 165-166. Not only surface buoyancy in North Atlantic but also Ekman pumping in the Southern Ocean.*

Responses: Thank you very much for point out this detail. In the revised manuscript, we changed this statement to “*since both the surface buoyancy in the North Atlantic and the Ekman pumping in the Southern Ocean are almost identical to those in Real (Figs. 4a4, 5a4).*”

4. *l. 176-177. This sentence reads a bit weirdly because the AM does not alter the moisture pattern either but still affect the MOC intensity via alteration of the atmosphere dynamics.*

Responses: Thank you very much for pointing out this problem. We should say that the presences of RM and GL do not significantly alter the global atmospheric moisture pattern and the tropical atmospheric circulation, so that they have very limited effects on the AMOC and PMOC.

If we agree that the AMOC or PMOC consists of the thermohaline component (mainly determined by the buoyancy flux in the mid-to-high latitudes) and the wind-driven component (mainly determined by the wind stress and Ekman pumping in the tropics), it should be easy to understand that the RM and GL’s minimal roles in the AMOC and PMOC. However, the AM affects the wind-driven component of the PMOC via altering the tropical wind system although it does not alter the moisture pattern in the mid-to-high latitudes either.

In the revised manuscript, we state that “*The presence of other large topographies, such as AM, RM, and GL, would not switch the MOC from the Pacific to the Atlantic.*” (in lines 178-180); and “*On the other hand, both RM and GL have minimal effects on the PMOC (Figs. 3a3-f3) because their presences do not significantly alter the global atmospheric moisture pattern and the*

atmospheric circulation in the tropics (Extended Data Fig. 6).” (in lines 183-186). Extended Data Fig. 6 is added to this revision. It is also added here as Fig. R1 for reference.

Fig. R1 Equilibrium changes in atmospheric circulation and moisture transport. (a1)-(a3) changes in vertically integrated moisture transport (vector; units: $\text{kg}\cdot\text{m}^{-1}\text{s}^{-1}$) and its convergence (shading; units: $10^{-5} \text{kg}\cdot\text{m}^{-2}\text{s}^{-1}$), (b1)-(b3) geopotential height (shading; units: 10 m) and winds (vector; units: m/s) at 850 hPa. (a1)-(a3) and (b1)-(b3) are changes in AM, RM and GL, respectively, with respect to Flat. The atmospheric moisture convergence (divergence) is plotted as positive (i.e., $-\nabla \cdot \vec{v}q > 0$) (negative, $-\nabla \cdot \vec{v}q < 0$), representing a gain (EMP<0) (loss, EMP>0) of ocean freshwater from (to) the atmosphere.

5. *Methods. Are changes in runoff consequent in the simulations? What would happen if the runoff was adequately re-routed?*

Responses: Thank you very much for these questions. The runoff is calculated in Common Land Model (CLM) of CESM1.0. It includes the liquid water runoff (R) and ice runoff (I). The changes of runoff in the simulations are adjusted according to the River Transport Model (RTM). The RTM uses a linear transport scheme at 0.5° resolution to route water from each grid cell to its downstream neighboring grid cell. The ocean freshwater liquid and ice fluxes are passed to the flux coupler that distributes the fluxes to the appropriate ocean grid cells.

In all experiments, the ice_runoff is set to “False” and the ice runoff is zero. In response to the topographic change, the direction and discharge of river runoff will be changed. In CESM1.0, the

drainage or sub-surface runoff is based on the SIMTOP scheme (Niu et al., 2005). Due to the limited geological data, it is difficult to get the exact distributions of river runoff during the period of mountain uplift. There is no artificial modification to the river runoff. The results of runoff changes are based on the model simulations.

Figure R2 show the runoff flux (units: mSv) anomalies in several experiments relative to Flat. Positive value means that the ocean gains fresh water. In general, the effect of river runoff change on the ocean circulation can be neglected.

Fig. R2 River runoff flux (units: mSv) anomalies in experiments TP, AT, TP+AT, and Real, with respect to Flat. Positive value means freshwater gain by the ocean.

Reference:

Niu, G.-Y., Yang, Z.-L., Dickinson, R.E., and Gulden, L.E. 2005. A simple TOPMODEL-based runoff parameterization (SIMTOP) for use in global climate models. 110: D21106. DOI: 10.1029/2005JD006111.

6. *Fig. 4. I get what dots and crosses represent on Fig. 4a1-a4 but not on Fig. 4b1-b4 (the difference figures). Do crosses (dots) represent increased downwelling (upwelling)?*

Responses: Black dots represent Ekman pumping, and crosses, Ekman downwelling (units: cm/day). In Fig. 4b1, black dots and crosses denote anomalous Ekman pumping and downwelling, respectively, in Flat, with respect to Real. In Figs. 4b2-b4, black dots and crosses denote Ekman pumping changes in Flat2Real, TP, and TP+AT, respectively, with respect to Flat. Figure 4’s caption was rewritten, and made clearer in the revised manuscript.

7. *Also what do you mean by “mean Ekman pumping/downwelling”? I do not understand how dots and crosses can have units of cm/day.*

Responses: The mean Ekman pumping/downwelling represents the Ekman pumping averaged over the last 100 years of each experiment. Sorry for the mistake in our previous manuscript. The dots and crosses do not have units, representing just the region where the upper ocean have positive (upward) and negative (downward) movements. The caption of Fig. 4 was rewritten, and made clearer in the revised manuscript.

8. *Fig. 5. Mass transport on Fig. 5b1-b4 should be in Sv to be consistent with the other figures.*

Responses: Thank you very much for this suggestion.

In the previous Figs. 5b1-b4, the mass transport across 30°S was obtained by zonal integration of meridional velocity, $V(z)*L_x$, which has the units of m^2/s .

Figures 5b1-b4 are replotted in the revised manuscript, in which the mass transport is calculated as $V(z)*L_x*dz(z)$, where $dz(z)$ is the layer depth of $V(z)$, so that the units are m^3/s (Sv). Now, the mass transport throughout the manuscript has consistent units.

9. *Fig. 6. What month or season is reflected in the sea-ice margin/velocity? It should be clearly written in the legend.*

Responses: Thank you very much for pointing out this problem. In Fig. 6a, the sea-ice coverage is the annual averaged, and the MLD is for March. In Fig. 6b, all changes are annual averaged. In the revised manuscript, these are clearly stated in figure captions.

10. *Extended data Fig. 1 should be plotted at model resolution because this is what matters in the results.*

Responses: Thank you for this suggestion. In the revised manuscript, Extended Data Fig. 1 is replotted with data at model resolution. Please refer to Fig. R3.

Fig. R3 Topography configurations in coupled model experiments.

References:

1. Elliot, D. H. (2013). The geological and tectonic evolution of the Transantarctic Mountains: a review. Geological Society, London, Special Publications, 381(1), 7-35.
2. Ferraccioli, F., et al. (2011). East Antarctic rifting triggers uplift of the Gamburtsev Mountains. *Nature*, 479(7373), 388-392.
3. Ferreira, D., et al. (2018). Atlantic-Pacific asymmetry in deep water formation. *Annual Review of Earth and Planetary Sciences*, 46, 327-352.
4. Sauermilch, I., et al. (2021). Gateway-driven weakening of ocean gyres leads to Southern Ocean cooling. *Nature communications*, 12(1), 6465.
5. Su, T., et al. (2019). Uplift, climate and biotic changes at the Eocene-Oligocene transition in south-eastern Tibet. *National Science Review*, 6(3), 495-504.
6. Xiong, Z., et al. (2020). The early Eocene rise of the Gonjo Basin, SE Tibet: From low desert to high forest. *Earth and Planetary Science Letters*, 543, 116312.

Response: These references are added in the revised manuscript.

Replies to Reviewer #2:

Thank you very much for these constructive comments. We have revised the manuscript carefully based on these suggestions. The following are our point-by-point replies.

The manuscript by Haijun Yang et al. presents an investigation into the influence of orography on the Meridional Overturning Circulation (MOC) using the low-resolution coupled model CESM. Previous studies, including Fallah et al. (2016), Su et al. (2018), and Yang and Wen (2020), have examined the effects of the Tibetan plateau on oceanic circulation. Maffre et al. (2017) conducted an analysis on the impact of global orography on oceanic circulation by comparing two simulations: one incorporating present-day geography and another using a flat Earth representation. In this regard, Haijun Yang et al.'s work combines these two approaches by investigating the effects of a flat Earth representation and various combinations of present-day orography on oceanic circulation. While this study is interesting, I have a few concerns regarding this approach.

Responses: Thank you very much for your valuable comments, which help us improve the manuscript tremendously. Considering the comments from all the reviewers, we revised the manuscript primarily in these following aspects:

- 1) Introduction to research background is much improved. We rewrote statements and make it clear to readers that there are many controversies regarding the on/off state of the AMOC/PMOC, as well as the timing of mountain uplift.
- 2) Mechanisms for global climate changes in response to mountain uplift are explained with more details;
- 3) Some relevant references are added; and most of figures are replotted.

Specific comments:

1. (L24-L39) *One concern with the sensitivity experiments conducted in this study is that they include the entire topography of the continent, rather than focusing solely on specific features like the Tibetan plateau or the Rocky Mountains. Previous research has demonstrated that the uplift of the Tibetan plateau influences the Atlantic Meridional Overturning Circulation (AMOC). However, by incorporating the entire topography of a continent, it becomes challenging to isolate the effects of individual landforms and determine whether minor features play a role in the establishment or disruption of the AMOC/PMOC.*

Responses: Thank you very much for this question.

In previous studies, we also conducted an experiment with only TP removed (Fig. R4a), while the Mongolian Plateau was unchanged. This experiment is called No_RegionalTP. The AMOC change in No_RegionalTP is almost identical to that in NoTibet (Fig. R4b), suggesting that the TP is important to the AMOC, while the Mongolian Plateau is not. However, for the PMOC, although the TP is the most important, the Mongolian Plateau also plays a role. This is qualitatively consistent with the finding of White et al. (2017), which disclosed an important role of the Mongolian Plateau in the Pacific wintertime atmospheric circulation. Since both the wind-driven and thermohaline dynamics are important to the PMOC establishment, the Mongolian Plateau can affect the wind-driven part of the PMOC through its role in the Pacific atmospheric circulation.

Fig. R4 (a) Topography configuration without Tibetan Plateau (60°-130°E, 20°-45°N) (No_RegionalTP). (b) Temporal evolutions of the PMOC (red) and AMOC (blue) in NoTibet (solid curves) and No_RegionalTP (dashed curves).

We also answered this question in our previous work on the TP's effect on the ENSO (Wen et al., 2020) (Fig. R5). The results from No_RegionalTP are nearly unchanged from those in NoTibet. $\sigma(\text{SST})$ in No_RegionalTP is roughly of the same magnitude as that in NoTibet (Fig. R6). The tropical SST anomaly shows much bigger oscillation after the TP removal (Fig. R6). Also, the mean ocean climate changes are very close to each other in No_RegionalTP and NoTibet: weakened trade winds, SST warming in the central-eastern Pacific, and SST cooling in the western Pacific, more freshwater gain in the central tropical Pacific, shallower MLD in the central tropical Pacific, and flattened thermocline (Fig. R7). In addition, we examined the atmospheric circulation change in No_RegionalTP (Fig. R8). We found that the atmospheric circulation changes over the tropical oceans are nearly identical in NoTibet and No_RegionalTP. Difference exists only in high latitudes. Our work also confirmed that the landform around the TP is much more important than that around the Mongolian Plateau in tropical climate change.

Fig. R5 Topography configuration in coupled model experiments. (a) is for the control simulation with realistic topography (Real), and (b) is for the experiment without the regional Tibetan Plateau (No_RegionalTP; 60°-140°E, 20°-45°N). Units: m.

Fig. R6 Time series of standard deviation of SST anomalies ($\sigma(\text{SST})$; °C) averaged in the Niño-3 region (150°-90°W, 5°S-5°N). The $\sigma(\text{SST})$ field is smoothed with a sliding window of 11 years. Black line is for Real, red is for NoTibet, and orange is for No_RegionalTP.

Fig. R7 Quasi-equilibrium changes in mean tropical climate, including (a): SST ($^{\circ}\text{C}$) and surface wind stress (dyn/cm^2), (b): precipitation minus evaporation (PmE; $10^{-5} \text{ kg}/\text{m}^2/\text{s}$), (c): MLD (m), and (d): thermocline depth (m) in NoTibet. (e-h) are the same as (a-d), except for No_RegionalTP.

Fig. R8 Quasi-equilibrium changes in (a) geopotential height (shading; m) and wind (vector; m/s) at 850 hPa and (b) vertically integrated water vapor transport ($\rho_a \vec{v}q$; vectors; $\text{kg}/\text{m}/\text{s}$) and its convergence ($-\rho_a \nabla \cdot (\vec{v}q)$; shading; $10^{-5} \text{ kg}/\text{m}^2/\text{s}$) in NoTibet. (c-d) are the same as (a-b), except for No_RegionalTP.

References:

- Wen, Q., K. Doos, Z. Lu, Z. Han, and H. Yang, 2020: Investigating the role of the Tibetan Plateau in ENSO variability. *J. Climate*, 33, doi: 10.1175/JCLI-D-19-0422.1.
- Sha, Y., Z. Shi, X. Liu, and Z. An, 2015: Distinct impacts of the Mongolian and Tibetan Plateaus on the evolution of the East Asian monsoon. *J. Geophys. Res.*, **120**, 4764–4782, doi:10.1002/2014JD022880.
- Shi, Z., X. Liu, Y. Liu, Y. Sha, and T. Xu, 2015: Impact of Mongolian Plateau versus 759 Tibetan Plateau on the westerly jet over North Pacific Ocean. *Climate Dyn.*, **44**, 3067–3076, doi:10.1007/s00382-014-2217-2.
- White, R. H., D. S., Battisti, and G. H. Roe, 2017: Mongolian mountains matter most: impacts of the latitude and height of Asian orography on Pacific wintertime atmosphere circulation. *J. Clim.*, **30**, 4065–4082.

2. *Additionally, the study includes the Andes Mountains and the entire South American, African, Middle Eastern regions up to the Zagros Mountains and Anatolian plateau under the scenario "AM". Regarding the ice sheet (Antarctica and Greenland), the study tests the sensitivity to changes in elevation, but it is not specified whether the authors maintained the albedo of ice when removing the Antarctica / Greenland topography (the term ice sheet is never employed in the manuscript). This information is crucial as altering the albedo would affect the energy balance and potentially influence the circulation patterns and precipitation.*

Responses: Thank you very much for pointing out these problems.

In Flat2Real, it's true that when adding AM after AT, we actually add the whole South American topography, the African topography, the Australian topography, and most of the European topography (Extended data Fig. 1d). However, in single topography experiment AM (Fig. R9a) and combined-topography experiment TP+AM (Fig. R9b), the AM only includes the South American topography.

Fig. R9 Topography configuration in coupled model experiments. (a) is for single AM experiment, and (b) is for the combined experiment TP+AM. Units: m.

As far as the AMOC and PMOC are concerned, the combined effect of the South American topography (without the AM), the African topography, the Middle Eastern regions up to the Zagros Mountains, and the Anatolian Plateau are insignificant, which can be deduced from Fig. 1 and Fig. 2a. In Fig. 1, the PMOC after adding AM (from year 2400 to year 3400) is about 20 Sv, which

includes the combined effect of those topographies. In Fig. 2a, the PMOC after adding AM to Flat is about 18 Sv, which does not include the combined effect of those topographies. The AMOC during years 2400-3400 of Fig. 1 is about 2 Sv, while it is about 1 Sv in only-AM experiment in Fig. 2a. Due to the limitation of computer resources, we did not design experiments to explicitly investigate the effects of the South American topography (without the AM), the African topography, the Middle Eastern regions up to the Zagros Mountains, and the Anatolian Plateau on the PMOC and AMOC.

Since this work only focuses on the topography effect, in all experiments we only modify the height of topography, keeping the albedo at the same value of Flat. The albedo can be freely adjusted with the integration of simulations. The dynamic ice sheet component in our experiments is closed. In the model setting, the land ice component is set to SGLC (stub glacier model), which means no dynamic ice sheets. The glacier areas and elevations are taken entirely from CLM's surface dataset; and no downscaling is done over non-glacier land units. The ice sheets are treated as big bright rocks. The bare ice albedo is prescribed to be 0.50 by default. The albedo of glacier is 0.80 for visible radiation.

In the revised manuscript, we explicitly state how the ice sheet is treated in the experiments. We agree that altering the albedo would affect the energy balance and potentially influence circulation patterns and precipitation patterns. However, in such a short manuscript, we cannot investigate albedo effect in detail. We would like to emphasize that the initial value of albedo over topography region is kept the same as that in FLAT, and it will self-adjusted according to thermal conditions during model integration. For example, with the uplift of a mountain, the albedo will increase.

In the 2nd paragraph of Methods section of the revised manuscript, we rewrote the related sentence as follows: *“Except for topography height, all other boundary conditions remain the same as in Flat and Real. The ocean-land configuration is set to modern-day conditions without correction for plate tectonic motion. Atmospheric CO₂ concentration is maintained at the preindustrial level (285 ppm). Changes in river routing and vegetation type are not considered. Continental ice sheets are treated as bright rocks in the model. The planetary albedo can adjust by itself according to thermal conditions. These experiments are conducted as single-variable (orography) sensitivity tests, rather than paleoclimate simulations involving multiple prescribed geologic boundary conditions.”*

3. *Moreover, when orography is removed, the drainage basin system is expected to undergo changes. The authors indicate that river routing is kept unchanged. Understanding the modifications in the drainage basin system is relevant to comprehending the overall impact of orography removal on oceanic circulation.*

Responses: Thank you very much for raising this question.

In the actual process of topographic uplift, the direction and discharge of river runoff will be changed. In CESM1.0, drainage or sub-surface runoff is based on the SIMTOP scheme (Niu et al., 2005). Due to limited geological data, it is difficult to get the exact distributions of river runoff during the period of mountains uplift. There is no artificial modification of river runoff. The results of runoff changes are based on model simulations.

Figure R10 shows the runoff flux (units: mSv) anomaly in several experiments (TP, AT, TP+AT, Real) relative to Flat. We can conclude that the anomalous freshwater flux due to the changes in the drainage basin system has a very limited impact on the global-scale ocean circulation.

Fig. R10 River runoff flux (units: mSv) anomaly in experiments TP, AT, TP+AT, and Real, with respect to Flat. Positive value means the ocean gain fresh water.

Reference:

Niu, G.-Y., Yang, Z.-L., Dickinson, R.E., and Gulden, L.E. 2005. A simple TOPMODEL-based runoff parameterization (SIMTOP) for use in global climate models. 110: D21106. DOI: 10.1029/2005JD006111.

- 4. These concerns regarding the specificity of landforms, albedo adjustments, and drainage basin modifications should be addressed in the manuscript to provide a comprehensive analysis of the study's findings.*

Responses: Thank you very much for this suggestion. In the revised manuscript, we added some detail on how we treat the changes in landforms, albedo, drainage basin, and river runoff flux in Method. Please refer to the replies to Q.2 above.

Due to space limited by the journal, a comprehensive analysis of these factors on ocean circulations is not feasible. We are working on a longer manuscript, in which comprehensive analysis will be conducted.

- 5. Despite the fact that the authors exclude to explain the evolution of AMOC during the Cenozoic and restrain this explanation to the role of Tibetan plateau in the present-day geography, the Figure 1 exhibits the impact of a sequential uplift of mountains with a “geological” chronology. The sequence is supposed to mimic the main phase of uplift (or growth of ice sheet). Although the authors do not try to provide a detailed explanation of the evolution of the AMOC/PMOC during the Cenozoic era and instead focus on the role of the topography in present-day geography, Figure 1 demonstrates the impact of sequential mountain uplift with a geological chronology. The sequence is intended to simulate the impact on ocean circulation of the primary phase of uplift or the growth of an ice sheet.*

Responses: Thank you very much for this comment. To some extent, we can learn from experiment Flat2Real about how the AMOC/PMOC could be established in the past. However, the actual evolution processes of the AMOC/PMOC during the Cenozoic era were much more complicated, because they involved so many boundary conditions. In a short manuscript like this one, we can only focus on a narrow topic, namely, the role of some giant topography in the AMOC in present-day climate.

- 6. The length of the simulation is a matter of debate. Following the addition of Antarctic, the simulation is integrated for 400 years. However, this duration may be considered relatively short to ensure that the deep ocean layers have fully reached equilibrium with the boundary conditions.*

Responses: Thank you very much for pointing out this problem.

We should have run +AT of Flat2Real for a much longer time (at least 1000 years). In the very early stage of this work, we did not realize this problem and just kept running the model, since the AMOC does not increase at this stage. We were just eager to see when the AMOC can get up.

To make up this problem, in the later stage of this project we ran the experiment OnlyAT from Flat for 1600 years (Fig. 2a) and found that the PMOC in OnlyAT could reach 18 Sv and the AMOC was still not established. Therefore, we can say that during years 2000-4000 of Flat2Real, the AT did contribute to the strong PMOC in the later stage; and at the same time, it did not lead to the AMOC formation in the later stage.

7. *(L69) The authors do not provide an explanation regarding whether the albedo was modified when removing the topography of Antarctica. It would be beneficial to clarify whether any adjustments were made to account for changes in albedo resulting from the modification of Antarctic topography.*

Responses: Thank you very much for raising this question. Please also refer to our replies to Q.2.

In all experiments, we only modify the height of topography, keeping the albedo at the same value of Flat. The albedo can freely adjust during the integration of the simulation. The dynamic ice sheet component in our experiments is turned off, which means no dynamic ice sheets. The ice sheets are treated as big bright rocks. The bare ice albedo is prescribed to be 0.50 by default. The albedo of glacier is 0.80 for visible radiation.

Figure R11 shows albedo in different experiments. The small difference over the Antarctic continental is from model's adjustments. It's not caused by changes in ice sheets. Albedo is calculated using $(FSDS-FSNS)/FSDS$, where FSDS represents downwelling solar flux at the surface, and FSNS represents net solar flux at the surface.

Fig. R11 Surface albedo in different experiments

In our previous study (Wen et al., 2021), we conducted a detailed study of albedo effects and found that changes in albedo over the TP (thermal effect) have a much smaller impact on the ocean circulation, when compared with TP's dynamic effect.

Reference:

Wen Q, Yang H, Yang K, et al. Possible thermal effect of Tibetan Plateau on the Atlantic meridional overturning circulation[J]. Geophysical Research Letters, 2022, 49(4): e2021GL095771.

8. *(L93-...)* The authors primarily focus on describing the oceanic changes in their study, driven by atmospheric modifications resulting from the addition or removal of topography. However, they do not provide a detailed explanation of the underlying mechanisms involved. It would indeed be valuable to include a few sentences elucidating the specific mechanisms by which the evaporation or precipitation patterns are altered when modifying the topography.

Responses: Thank you very much for this suggestion.

The underlying mechanisms are detailed in the following paragraph (Lines 108-148). In our previous studies (Fig. 4 in Wen and Yang, 2020; Fig. 12 in Yang and Wen, 2020), we discussed the mechanisms thoroughly. Removing the TP leads to enhanced atmospheric water vapor transport from the Pacific to the Atlantic, resulting in decreased freshwater flux over the western Pacific and increased freshwater flux over the North Atlantic, which lead to the PMOC formation and AMOC decline.

9. *In this work, the authors do not explicitly mention conducting statistical significance tests to evaluate the differences between two variables.*

Responses: Thank you very much for pointing out this problem.

We have done statistical tests for all variables, but did not mention them in this manuscript explicitly due to the limitation of word count.

Most changes seen in the experiment results are statistically significant. We did Mann-Kendall test for all figures related to the differences between two variables. For the clarity of visual effect, we decided not to show the significance test in the figures.

In the last paragraph of Methods section of the revised manuscript, we state that *“The student-t test is used to examine the statistical significance of our results. Most changes are significant at the 95% confidence level, which is expected because altering large topography induces strong mechanical forcing and obvious responses around the globe. For clarity, significance test is not provided in any figures.”*

Figure R12 shows the SSS change in TP, AT, TP+AT, and Real, with respect to FLAT, in which the crosses represent the area that exceeds the 95% significance level.

Fig. R12 Changes in annual mean SSS in TP, AT, TP+AT, and Real, with respect to FLAT. Stippling indicates changes exceeding the 95% significance level according to the Mann-Kendall trend test.

10. (L112-122) *The mechanism causing a change in precipitation or evaporation (or both) are discussed. The changes in atmospheric circulation and moisture transport are associated to changes in GPH. The authors do not explicitly delve into the topic of temperature changes in relation to GPH variations, nor do they discuss the potential role of Sea Surface Temperature (SST) changes in the North Atlantic affecting the North Atlantic Deep Water (NADW) circulation. To provide a more comprehensive analysis, it would be beneficial for the authors to address the possible temperature changes resulting from GPH modifications and discuss the influence of SST changes on NADW formation explicitly. Incorporating these aspects would further elucidate the connections between atmospheric circulation, temperature, and oceanic processes, enhancing the overall understanding of the research findings.*

Responses: Thank you very much for raising this question.

In this work, we state that “*The change in SSD from Real to Flat can be primarily attributed to the change in SSS*” and did not discuss the SST change in the North Atlantic, because we had a thorough discussion on this topic in our previous papers (e.g., Yang and Wen, 2020). Due to the limitation of paper length, in the revised manuscript we added a brief statement that the temperature effect on the AMOC is not important.

Figure R13 (i.e., Fig. 2 of Yang and Wen, 2020) shows that after 50 years of the TP removal, the AMOC exhibits a roughly linear decline for about 200 years. The AMOC is finally weakened by more than 80% and practically shut down in about 300 years.

As the first step to understand the AMOC change, the temporal evolution of surface buoyancy change averaged in the NADW region is examined in Fig. R13b. The sea-surface density (SSD) increases during the first 50 years and then decreases linearly (black curve), consistent with the change of the AMOC index (Fig. R13a). The SSD change consists of SST-induced change (dashed red) and sea-surface salinity (SSS)-induced change (dashed blue). Note that the surface ocean keeps cooling in the whole 400 years (red curve), which always increases the SSD (dashed red). In contrast, the SSS increases first and then decreases (blue curve), followed closely by the SSD change (dashed blue and black curves). Based on Fig. R13b, we quantify that the increase of SSD during stage I is contributed by surface cooling (30%) and surface salinization (70%), while the SSD decrease later on is totally contributed by surface freshening. It clearly illustrates that the weakening of the AMOC should be attributed to the freshwater increase in the NADW region

Fig. R13 [REDACTED]

Figure R14 (i.e., Fig. 4 in Yang and Wen, 2020) shows the horizontal patterns of buoyancy change in the North Atlantic in both stages. In stage I, the SST change has a tripolar structure (Fig. R14a): a significant cooling (more than 2°C) occurs in the midlatitudes between 40° and 60°N , saddled by two warming regions located in the GIN seas and tropics, respectively. The SSS change has an east-west dipole structure (Fig. R14b), with significant salinization in the Labrador Sea and south of the Greenland Sea and a weak freshening in the eastern North Atlantic. The combined effect of changes in SST and SSS results in SSD increase (Fig. R14c), particularly in the NADW region; this increases the MLD and thus the deep-water formation, leading to a stronger AMOC in stage I. In stage II, the quasi-equilibrium changes in SST, SSS, and SSD show rather simple structures (Figs. R14d-f), that is, significant cooling, freshening, and thus a lighter surface ocean in the entire North Atlantic, consistent with the “off” state of the AMOC. It is clear that the SSD changes in the North Atlantic in both stages are mostly determined by SSS change.

Fig. R14 [REDACTED]

Reference:

Yang, H., and Q. Wen, 2020: Investigating the role of the Tibetan Plateau in the formation of Atlantic meridional overturning circulation. *J. Climate*, 33(9), 3585-3601, doi: 10.1175/JCLI-D-19-0205.1.

11. (L122-L123) *“The expansion of sea ice (···) carries a substantial amount of freshwater ” : the sentence is not clear to me.*

Responses: Thank you very much for this question.

During the southward expansion of sea ice, there is a remarkable amount of sea-ice melting at the same time, which provides fresh water to the ocean, leading to the AMOC shutdown. In the revised manuscript, we revised this sentence and made it clearer in lines 129-131.

12. (L135) *“The absence of the Antarctic topography leads to an 80% decrease in wind stress”: is it due to temperature gradient?*

Responses: Thank you very much for this question.

This is mainly due to the change of meridional pressure gradient. Absence of the Antarctic topography causes an anomalous high pressure over the Antarctic, which produces an anomalous

northward pressure gradient, and thus an anomalous easterlies based on the geostrophic balance. In the revised manuscript, we made this point clearer in lines 141-144.

13. *Figure 5: the Ekman pumping/downwelling is discussed with the 850hPa wind stress. This is not clear. Why don't the authors show the wind stress at the sea surface?*

Responses: Thank you very much for raising this question.

In Fig. 5, the Ekman pumping/downwelling is calculated by using surface wind stress: $\omega_E = \text{curl}\left(\frac{\tau}{\rho_f}\right)$, where τ is surface wind stress. The wind field we used is at 850 hPa, because at this level the geostrophic balance is satisfied, so that the wind field can also represent the pressure field.

Figure R15 shows Ekman pumping and wind stress at sea-surface level. We can see that the ageostrophic component of the wind is strong, compared to that in Fig. 5. For the Ekman pumping, both geostrophic and ageostrophic components of the wind are important to produce vertical movement in the upper ocean.

Fig. R15 Surface wind stress (units: dyn/cm²) and Ekman pumping (units: cm/day). The values less than 0.015 dyn/cm² are not shown.

14. *Positive mass transport across 30°S is counted positively northwards. This is not mentioned in the figure caption.*

Responses: Thank you very much for this suggestion. We revised it in Fig. 5.

15. *(L202+Extended Data Figure 3a1-4) The authors' conclusion that the Tibetan plateau acts as a significant attractor of freshwater in the Northern Hemisphere (NH) is potentially accurate if the boundary conditions used in their scenario are limited to the Tibetan plateau alone. However, it is important to consider that changes in the North Pacific region can also be influenced by the westerlies blowing from Eastern Asia, which may be impacted by the presence or absence of landforms located to the north of the Tibetan plateau.*

Responses: Thank you very much for raising this question.

Please refer to our reply to Q.1 and Figs. R4, R5 and R8. It is true that changes in the North Pacific can also be influenced by the westerlies from East Asia, which can be impacted by the presence or absence of landforms (such like the Mongolian Plateau) to the north of the TP. However, the TP's impact dominates; so we did not separately consider the impact from the Mongolian Plateau.

16. *(L224) I think that the authors could quote a more recent paper than the Ruddiman's 1989 paper*

Responses: Thank you very much for this suggestion. We added a recent review paper published in *Nature* as reference #46 in the revised manuscript.

Wu, F., Fang, X., Yang, Y. *et al.* Reorganization of Asian climate in relation to Tibetan Plateau uplift. *Nat Rev Earth Environ* **3**, 684–700 (2022).

17. *(L232-243) I agree with the observation regarding the limitations of the study, particularly with respect to the spatial resolution of the atmospheric model. I wonder whether the low resolution of the model might not be a potential factor contributing to the lack of impact observed for certain mountain ranges, particularly those with an elongated shape such as the Andes or the Rockies. Mountain ranges with complex topography may not be adequately captured or represented in the model due to the coarse resolution. In contrast, the Antarctic ice cap and the Tibetan plateau may be better resolved within the model. This could explain why their impacts are more apparent in the simulations.*

Responses: Thank you very much for these insightful thoughts.

We are going to repeat some experiments using high-resolution version of CESM2.0. This will be very resource-consuming and take a long time. We hope to have some new fascinating findings.

Other comments:

1. *(L24-33) The timing of uplift is also debatable. A part of the Tibetan plateau is largely uplifted before the Late Miocene.*

Responses: Thank you very much for this comment. We mentioned this problem in the revised manuscript in lines 38-40.

2. *Antarctic (ice sheet?): the authors explains that “Antarctic rapidly expanded in the Oligocene and persisted until the Late Oligocene”. This is not clear to me what the authors mean by “persisted until the Late Oligocene”. Does it mean that the ice sheet melts after the Late Oligocene ? No reference is provided.*

Responses: Thank you very much for raising this question. Here, we mean the glaciation of the Antarctic persists from the early Oligocene to the late Oligocene (34-26 Ma). In the revised manuscript, we rewrote this sentence as follows: “...*Although the transantarctic and Gamburtsev Mountains over the Antarctica were likely already present at the start of the Cenozoic (65 Ma) (Ferraccioli 2011; Elliot 2014), the glaciation of the Antarctica was thought to occur during the Eocene-Oligocene (56-23 Ma)...*”

3. *Andes uplift => reference 19 : Zachos et al. 2001 ?* Revised
4. *Figure 5: continent contours are too thin to be seen when printed* Replotted.
5. *Extended Data Figure 3: continent contours are hard to see* Replotted.

REVIEWER COMMENTS

Reviewer #1 (Remarks to the Author):

This is the second review of “Which Mountains Matter to Global Meridional Overturning Circulation” by Haijun Yang and coauthors for potential publication in Nature Communications and I first apologize for taking so long to complete this review.

In my first set of comments, I recommended the manuscript to be returned to the authors with minor revisions. The experimental setup was sound and the results generally robustly demonstrated but the context in which the study is integrated was not very well described and the relevance of the findings to the understanding of the MOC evolution remained unclear.

I do not think the manuscript has been much improved from this perspective.

To be more specific, I still think that the core of the paper, that is, the physical results, is a valuable addition to the literature on the role of mountains in the climate system, and I would like to see these results published. However, the Discussion and, to a lesser extent, the Introduction, are still not up to the job, in particular for publication in a journal like Nature Communications.

The Introduction, which mostly reviews the Cenozoic evolution of the AMOC/PMOC and of the mountain ranges that are investigated later in the manuscript (l. 66-99), has been slightly expanded and reads better, though some errors remain (see below). My issue in this section is mainly with the last paragraph, which needs to be re-written to 1) state more clearly why the authors chose to keep the modern land-sea mask and 2) not elude the fact that your Flat2Real simulation is an attempt of simulating the Cenozoic AMOC/PMOC evolution.

Something along these lines for example: “Here, as a first step, we numerically investigate how the presence or absence of various major mountain ranges affect the GMOC as well as their cumulative impact when they are uplifted sequentially in the model in an initial flat Earth simulation. In order to isolate the topographic effects from plate tectonics and long-term bathymetric and atmospheric changes, we keep the modern bathymetry and continental positions, as well as modern greenhouse gas concentrations, incident solar radiation and orbital parameters. We show that, in our simulations, the TP uplift is the primary driver of the shutdown of a PMOC and initiation of an AMOC, although high Antarctic topography is required to drive a strong, modern-like, AMOC. We further discuss the implications of our results to the long-term Cenozoic history of the AMOC/PMOC.”

The Discussion is perhaps more problematic, as it currently stands, and I note that it has only barely been revised, in spite of comments from both reviewers. The work presented here may well be “just a beginning” as the authors claim in their rebuttal, yet this must not preclude them to better discuss their results.

Here are a couple of comments to improve it:

1) the 3rd paragraph (l. 289-300) reads quite weird. It starts with “rather than mimicking past climate evolution” but the next sentence ends with “The results [...] provide insight into [...] the climate

development in the Late Cenozoic". A few lines later "this study establishes the direction of MOC change as a function of increasing uplift over time". What is the goal of the study in this case? I do not see it problematic that the authors attempt to interpret their results in light of long-term Cenozoic climate change but it needs to be discussed: what are the implications of a modern land-sea mask? And of modern gateways and/or greenhouse gas?

2) Just a single line stating that the authors "did not consider the effects of continental drift and oceanic gateway switches" is just simply not enough, in particular as the interpretation of the results as consistent with the general direction of Cenozoic climate change is repeated several times in the Discussion (see comment above and again l. 316).

3) The list of other studies referenced in the last paragraph should be better integrated. For example, the authors state that "Maroon discovered that the RM can have a significant effect on GMOC through its impact on hydrology", which is opposite to what the authors found in their experiments (l. 242-245). Why?

Specific comments in the Introduction:

l. 68-70. Worth integrating DeepMIP results? Most models do not simulate an AMOC or PMOC in the Early Eocene (Zhang et al. 2022).

l. 70-71. "during the Early Oligocene period (about 35-33 Ma)" => "at a later stage".
By the way, the 35-34 Ma interval is still Eocene.

l. 77. "the former has a net evaporation" => "the former is a net evaporative basin".

l. 81. "is believed" => "is also believed"

l. 82. Incorrect reference. Ref. 12 does not discuss the impact of Drake and Tasman gateways on the AMOC. Could cite Hutchinson et al. (2019) instead.

l. 84-85. "can precipitate contrasting shifts in the global MOC" => "holds the potential to trigger large-scale transitions in the global MOC".

l. 87 and throughout. "over the Antarctica" => "over Antarctica" or "over the Antarctic continent".

l. 88-89. "the glaciation of the Antarctica is believed to have occurred during the Eocene-Oligocene (56-23 Ma)." This is very vague. The consensus today is that the first large-scale glaciation of Antarctica occurred at the EO transition (34-33.5 Ma).

l. 90. Incorrect reference. Ref. 20 is not about the Andes. I suggest that the authors use the Boschmann (2021) paper as their new reference on the uplift of the Andes, and revise their chronology in consequence.

I. 93-94. “the rapid and main uplift of the TP was realized between 10 and 8 Ma”. Not sure this is correct. At least two of the references (25-29) are > 30 years old and they should be updated, given the dynamism existing about the TP uplift history. Latest reviews suggest that only small parts of the TP reached their modern elevation after the Early (20 Ma) or Mid (15 Ma) Miocene. See, e.g., Tardif et al. (2023).

In the end, I think that the revisions required are still quite minor, but necessary, and I trust the authors to adequately revise their manuscript.

References

Boschman, L. M. (2021). Andean mountain building since the Late Cretaceous: A paleoelevation reconstruction. *Earth-Science Reviews*, 220, 103640. <https://doi.org/10.1016/j.earscirev.2021.103640>

Hutchinson, D. K., Coxall, H. K., O’Regan, M., Nilsson, J., Caballero, R., & de Boer, A. M. (2019). Arctic closure as a trigger for Atlantic overturning at the Eocene-Oligocene Transition. *Nature Communications*, 10(1), 3797. <https://doi.org/10.1038/s41467-019-11828-z>

Tardif, D., Sarr, A. C., Fluteau, F., Licht, A., Kaya, M., Ladant, J. B., ... & Banfield, W. (2023). The role of paleogeography in Asian monsoon evolution: Associated a review and new insights from climate modelling. *Earth-Science Reviews*, 104464. <https://doi.org/10.1016/j.earscirev.2023.104464>

Zhang, Y., de Boer, A. M., Lunt, D. J., Hutchinson, D. K., Ross, P., van de Flierdt, T., ... & Huber, M. (2022). Early Eocene ocean meridional overturning circulation: The roles of atmospheric forcing and strait geometry. *Paleoceanography and Paleoclimatology*, 37(3), e2021PA004329. <https://doi.org/10.1029/2021PA004329>

Reviewer #2 (Remarks to the Author):

The manuscript by Haijun Yang et al. offers a comprehensive description of the physical mechanisms influenced by the presence / absence of large landform systems. The article is commendably clear and well-written. The responses to the reviewers' inquiries provide valuable insights that enhance our understanding of the authors' methodology.

The authors appropriately acknowledge the study's limitations in the conclusion, notably highlighting factors such as the palaeogeography anchored in the present and the absence of atmospheric CO₂ variation, preventing direct comparison with the evolution of Cenozoic climates. I fully concur with this assessment.

However, in the conclusion, certain sentences imply that Tibet played a dominant role in the evolution of deep-water formation. To enhance clarity, it is advisable to ensure that the key points and the summary explicitly convey the nuanced nature of the study's findings and the limitations acknowledged in the conclusion. This adjustment will contribute to a more precise and comprehensive understanding of the research presented.

Key 1 : The authors contrast their results with previous studies, but the conditions of the Jones and Cecci (2017) study are different, so the comparison seems risky to me.

L84: add more recent references

L89: The authors are encouraged to update the age of glacial inception in Antarctica, recognizing that the onset of glaciation in Antarctica is typically dated to the Late Eocene. This timing is attributed to a CO₂ atmospheric level that was considered too high before the Late Eocene. Please consider incorporating a relevant reference to support this updated information, providing a solid foundation for the revised age of glacial inception in Antarctica.

L90: Change the reference.

L83-99 : The paragraph in question lacks clarity as the authors attempt to draw a comparison between the age of Antarctic glaciation and the age of the Andes uplift. It is recommended that the authors revisit and revise this paragraph to articulate the comparison more explicitly, providing a clearer and more coherent presentation of the relationship between the two geological events.

Leier et al. 2013 : First uplift Early Miocene, second uplift Late Miocene – Andes Bolivia

L142-151 : The manuscript does not thoroughly discuss the intensification of the Pacific Meridional Overturning Circulation (PMOC) in relation to the uplift of the Andes

Figure 4 -Line 8 : Does the case Flat2Real correspond to the end of the experiment when all reliefs are uplifted (year 5601-6000) ?

L201: During the summer in Antarctica, the flat topography of the continent is likely to promote convection. Additionally, the presence of a high-pressure anomaly can indeed be seasonal.

Figure 5a1 : the figure 5a1 represents the difference Real minus Flat (on the plot) or Flat2Real minus flat (in the caption)

L246-262 : The manuscript describes the retreat of sea ice in the subpolar Atlantic, but the underlying cause of this retreat is not clearly explained. It is crucial to address this gap by providing a detailed and explicit discussion on the factors driving the retreat of sea ice in the subpolar Atlantic.

L289-291 : the authors explained clearly on lines 289-291 that this study is based on sensitivity

experiments (presence or absence of relief) and does not mimic past climate evolution. I fully agree with that. I regret that the key points and the abstract are less explicit.

Then the authors compare with their results with the climate evolution during the Cenozoic. I disagree with the sentence (L300) suggesting that "this study establishes the direction of MOC change as a function of increasing uplift over time". The position of continents (and relief) remains unchanged, as are the gateways (these limitations are indicated on L307).

The clarification provided in lines 289-291 regarding the study's reliance on sensitivity experiments rather than a direct mimicry of past climate evolution is appreciated. However, there seems to be a discrepancy in the key points and abstract, where the explicitness might be lacking.

Concerning the comparison with the climate evolution during the Cenozoic, there is disagreement with the sentence on line 300 suggesting that " this study establishes the direction of MOC change as a function of increasing uplift over time ". The disagreement arises from the understanding that the positions of continents (including relief) and gateways remain unchanged, as stated on line 307, and this should be reflected in the conclusions. A careful review and adjustment of these statements in the key points and abstract will contribute to a more accurate representation of the study's findings and limitations.

Replies to Reviewer #1:

Thank you very much for giving us the second chance to improve this manuscript. We have revised the manuscript carefully based on your constructive suggestions. The following are our point-by-point replies.

This is the second review and I first apologize for taking so long to complete this review.

In my first set of comments, I recommended the manuscript to be returned to the authors with minor revisions. The experimental setup was sound and the results generally robustly demonstrated but the context in which the study is integrated was not very well described and the relevance of the findings to the understanding of the MOC evolution remained unclear.

I do not think the manuscript has been much improved from this perspective.

To be more specific, I still think that the core of the paper, that is, the physical results, is a valuable addition to the literature on the role of mountains in the climate system, and I would like to see these results published. However, the Discussion and, to a lesser extent, the Introduction, are still not up to the job, in particular for publication in a journal like Nature Communications.

The Introduction, which mostly reviews the Cenozoic evolution of the AMOC/PMOC and of the mountain ranges that are investigated later in the manuscript (l. 66-99), has been slightly expanded and reads better, though some errors remain (see below). My issue in this section is mainly with the last paragraph, which needs to be re-written to 1) state more clearly why the authors chose to keep the modern land-sea mask and 2) not elude the fact that your Flat2Real simulation is an attempt of simulating the Cenozoic AMOC/PMOC evolution.

Something along these lines for example: “Here, as a first step, we numerically investigate how the presence or absence of various major mountain ranges affect the GMOC as well as their cumulative impact when they are uplifted sequentially in the model in an initial flat Earth simulation. In order to isolate the topographic effects from plate tectonics and long-term bathymetric and atmospheric changes, we keep the modern bathymetry and continental positions, as well as modern greenhouse gas concentrations, incident solar radiation and orbital parameters. We show that, in our simulations, the TP uplift is the primary driver of the shutdown of a PMOC and initiation of an AMOC, although high Antarctic topography is required to drive a strong, modern-like, AMOC. We further discuss the implications of our results to the long-term Cenozoic history of the AMOC/PMOC.”

Responses: Thank you very much for your valuable comments. The last paragraph of the introduction is revised following your suggestion (see below). Please also refer to lines 45-53 of the revised text.

“Here, as a first step, we numerically investigate how the presence or absence of various major mountain ranges affect the GMOC as well as their cumulative impact when they are uplifted sequentially in the model from an initial flat Earth. In order to isolate the topographic effects from plate tectonics and long-term bathymetric and atmospheric changes, we keep the modern bathymetry and continental positions, as well as modern greenhouse gas concentrations, incident solar radiation and orbital parameters. We show that, in our simulations (Methods and Table 1), the TP uplift is the primary driver of the PMOC shutdown and the AMOC initiation, although high Antarctic topography is required to drive a strong, modern-like AMOC. We further discuss the implications of our results to the long-term Cenozoic history of the AMOC/PMOC.”

The Discussion is perhaps more problematic, as it currently stands, and I note that it has only barely been revised, in spite of comments from both reviewers. The work presented here may well be “just a beginning” as the authors claim in their rebuttal, yet this must not preclude them to better discuss their results.

Here are a couple of comments to improve it:

1) the 3rd paragraph (l. 289-300) reads quite weird. It starts with “rather than mimicking past climate evolution” but the next sentence ends with “The results [···] provide insight into [···] the climate development in the Late Cenozoic”. A few lines later “this study establishes the direction of MOC change as a function of increasing uplift over time”. What is the goal of the study in this case? I do not see it problematic that the authors attempt to interpret their results in light of long-term Cenozoic climate change but it needs to be discussed: what are the implications of a modern land-sea mask? And of modern gateways and/or greenhouse gas?

Responses: Thank you very much for your valuable comments. This paragraph is rewritten based on your suggestion (see below). Please also refer to lines 248-264 of the revised text.

“This research marks the beginning of our investigation into the effects of the presence or absence of various major mountain ranges on the GMOC, as well as their cumulative impact under modern conditions. Our research aims to specifically discern the impact of topography on ocean circulation from other contributing factors, by incorporating modern bathymetry and continental configurations, current greenhouse gas levels, incident solar radiation, and orbital parameters in our experimental setup. The individual mountain uplift experiments in this study allow us to explore the linkages between uplift regions and climate changes in remote areas of the modern world. The sequential mountain uplift experiment Flat2Real provides insight into specific periods of paleoclimate, such as the climate development in the late Cenozoic. Previous

studies suggested that the TP uplift played a key role in shaping Cenozoic climate through circulation changes and weathering^{13,49,50}: over the past 40 million years, this uplift led to substantial deflection of the atmospheric jet stream, intensified monsoonal circulation, increased rainfall on the front slopes of the Himalayas, and conducive conditions for the formation of deep and intermediate waters in the North Atlantic. By using experiment Flat as a starting point and tracing the changes due to various uplifts, we can quantify the linkages between uplifts and climate changes and understand how the MOC changes as the progressive uplift of continental mountains.”

The implications of a modern land-sea mask, ocean gateways and greenhouse gas in our experiments implies that the TP impact on the GMOC could be exaggerated. We have added the discussion in the following paragraph.

2) Just a single line stating that the authors “did not consider the effects of continental drift and oceanic gateway switches” is just simply not enough, in particular as the interpretation of the results as consistent with the general direction of Cenozoic climate change is repeated several times in the Discussion (see comment above and again l. 316).

Responses: Thank you very much for your valuable comments. In this paragraph, more discussions are added (see below). Please also refer to lines 270-282 of the revised text

“Additionally, we did not consider the effects of continental drift and oceanic gateway switches, nor did we treat Greenland and Antarctic glaciation dynamically. *The setup of modern land-sea mask and ocean gateways in our experiments implies that the TP impact on the GMOC could be exaggerated. After all, the Tethys Seaway closing, the Panama Isthmus closing and the Bering Strait opening all occurred after the TP uplift⁸, all of which may lead to significant increases in the continental ice sheets in both hemispheres and enhance the NADW formation⁵¹⁻⁵³.* Furthermore, the background climate in the experiments uses the preindustrial conditions with constant CO₂, and the effect of chemical erosion in rapidly uplifted areas on atmospheric CO₂ is not accounted for. Typically, atmospheric CO₂ levels cannot remain in a steady state during the time of intense tectonism, which can last for millions of years, due to the temperature-weathering feedback mechanism. *The TP uplift may have resulted in a decrease of atmospheric CO₂ level, which can also enhance the NADW formation due to the growth of large continental ice sheets in the NH⁴⁹.*”

3) The list of other studies referenced in the last paragraph should be better integrated. For example, the authors state that “Maroon discovered that the RM can have a significant effect on GMOC through its impact on hydrology”, which is opposite to what the authors found in their experiments (l. 242-245). Why?

Responses: Thank you very much for this suggestion. In the revised manuscript, we added several new references and more analyses in several places. Due to the limitation of paper length and references required by the journal, we deleted these lines and three references. However, here we would like to provide an answer to why Maroon's finding is different from ours.

In Maroon (2016), removing only RM leads to a significant weakening of the AMOC by about 30%, however, removing the orography globally only leads to the AMOC weakened slightly. The latter result is not consistent with results of Schmittner et al. (2011), Sinha et al. (2012) and our study.

Maroon herself stated in her PhD. thesis that *“The AMOC collapses due to the removal of global orography in the models of Schmittner et al. (2011) and Sinha et al. (2012), a result which our simulations do not produce. We performed an additional simulation (not shown) in which we removed orography globally and found that the AMOC weakened slightly ...”*. *“The AMOC in our model is not as sensitive to orography as in the models used in other studies. One possible explanation for the weak response of the AMOC in CM2Mc is that the region of net freshening ($P-E > 0$) extends too far south in our Control simulation when compared to the freshwater input from observed storm track. This bias likely originates from the low resolution of the atmospheric model (Brayshaw et al., 2009) and is reflected in the SST biases in the North Atlantic”*.

Maroon also stated that *“Warren (1983) argued that the North Pacific does not have a PMOC because $P - E > 0$. Following his argument, the freshwater biases in the North Atlantic of the Control simulation should make the North Atlantic more similar to the North Pacific and prevent an AMOC. And yet, there is an AMOC in the Control simulation, and it is relatively insensitive to the removal of the Rocky Mountains. These results suggest that the Rockies are not the primary reason why there is an AMOC.”*

Specific comments in the Introduction:

1. 1. 68-70. Worth integrating DeepMIP results? Most models do not simulate an AMOC or PMOC in the Early Eocene (Zhang et al. 2022).

Responses: Thank you very much for this suggestion. The results from DeepMIP project are very interesting and provide us with new insight.

In the revised manuscript, we removed the old reference 5 and cited Zhang et al. (2022) as the reference 8. In lines 13-16 of the revised text, we state that “a recent study from DeepMIP project

found that neither model results nor proxy data suggest NADW formation during the early Eocene, while the evidence for NPDW formation remains inconclusive⁸.”

2. l. 70-71. “during the Early Oligocene period (about 35-33 Ma)” => “at a later stage”. By the way, the 35-34 Ma interval is still Eocene. Revised as suggested.
3. l. 77. “the former has a net evaporation” => “the former is a net evaporative basin”. Revised
4. l. 81. “is believed” => “is also believed” Revised
5. l. 82. Incorrect reference. Ref. 12 does not discuss the impact of Drake and Tasman gateways on the AMOC. Could cite Hutchinson et al. (2019) instead.

Responses: Thank you very much for pointing out this mistake. The reference is changed to Hutchinson et al. (2019).

6. l. 84-85. “can precipitate contrasting shifts in the global MOC” => “holds the potential to trigger large-scale transitions in the global MOC”. Revised.
7. l. 87 and throughout. “over the Antarctica” => “over Antarctica” or “over the Antarctic continent”. Revised.
8. l. 88-89. “the glaciation of the Antarctica is believed to have occurred during the Eocene-Oligocene (56-23 Ma).” This is very vague. The consensus today is that the first large-scale glaciation of Antarctica occurred at the EO transition (34-33.5 Ma).

Responses: Thank you very much for this suggestion.

In lines 31-32 of the revised text, this statement is changed to “... the first large-scale glaciation of Antarctica is believed to have occurred during the Eocene-Oligocene transition period (34-33.5 Ma)^{20,21}...”.

We added two relevant references as ref. 20, 21 in the revised manuscript:

20. Jamieson, S. S. R., Sugden, D. E., & Hulton, N. R. J. The evolution of the sub-glacial landscape of Antarctica. *Earth and Planetary Science Letters* **293**, 1-27 (2010).
21. Jamieson, S. S. R., Ross, N., Paxman, G. J. G., Clubb, F. J. An ancient river landscape preserved beneath the East Antarctic Ice Sheet. *Nat Commun* **14**, 6507 (2023).

9. l. 90. *Incorrect reference. Ref. 20 is not about the Andes. I suggest that the authors use the Boschmann (2021) paper as their new reference on the uplift of the Andes, and revise their chronology in consequence.*

Responses: Thank you very much for pointing out this mistake.

The reference 20 is changed to Boschmann (2021). In lines 32-33 of the revised text, the statement about AMs is changed to “The uplift of Andes Mountains (AMs) started in the Late Cretaceous (~70 Ma)²² and matured around 15-10 Ma^{23,24}.”

10. l. 93-94. *“the rapid and main uplift of the TP was realized between 10 and 8 Ma”. Not sure this is correct. At least two of the references (25-29) are > 30 years old and they should be updated, given the dynamism existing about the TP uplift history. Latest reviews suggest that only small parts of the TP reached their modern elevation after the Early (20 Ma) or Mid (15 Ma) Miocene. See, e.g., Tardif et al. (2023).*

Responses: Thank you very much for this comment. We rewrote the statement on the TP based on your suggestion. Please also refer to lines 35-44 of the revised text.

“The timeline for the formation of the Tibetan Plateau (TP) is a topic of highly debate. Some studies argue that parts of the TP were already in place during the late Eocene (38-33 Ma)^{26,27}, while other research suggest that the TP's rapid and main uplift occurred between 10 and 8 Ma²⁸⁻³¹. A more recent study proposes that most of the TP had attained its current elevation before the Mid-Miocene (15 Ma)³². This timing coincides with the onset of NADW formation, suggesting a possible link between the TP uplift and the development of NADW. Recent research also indicates that the TP is a critical factor affecting changes in the GMOC^{33,34}. Nevertheless, it is important to recognize that the chronology of the uplift of major mountain ranges remains a subject of intense discussion and investigation.”

Here we removed the old reference 26 and added Tardif et al (2023) as reference 32.

In the end, I think that the revisions required are still quite minor, but necessary, and I trust the authors to adequately revise their manuscript.

Responses: Thank you very much for your invaluable comments, which help us to improve this manuscript greatly. All 4 references you provided are cited in the revised manuscript.

References

1. Boschman, L. M. (2021). Andean mountain building since the Late Cretaceous: A paleoelevation reconstruction. *Earth-Science Reviews*, 220, 103640. <https://doi.org/10.1016/j.earscirev.2021.103640>
2. Hutchinson, D. K., Coxall, H. K., O'Regan, M., Nilsson, J., Caballero, R., & de Boer, A. M. (2019). Arctic closure as a trigger for Atlantic overturning at the Eocene-Oligocene Transition. *Nature Communications*, 10(1), 3797. <https://doi.org/10.1038/s41467-019-11828-z>
3. Tardif, D., Sarr, A. C., Fluteau, F., Licht, A., Kaya, M., Ladant, J. B., ... & Banfield, W. (2023). The role of paleogeography in Asian monsoon evolution: Associated a review and new insights from climate modelling. *Earth-Science Reviews*, 104464. <https://doi.org/10.1016/j.earscirev.2023.104464>
4. Zhang, Y., de Boer, A. M., Lunt, D. J., Hutchinson, D. K., Ross, P., van de Flierdt, T., ... & Huber, M. (2022). Early Eocene ocean meridional overturning circulation: The roles of atmospheric forcing and strait geometry. *Paleoceanography and Paleoclimatology*, 37(3), e2021PA004329. <https://doi.org/10.1029/2021PA004329>

Replies to Reviewer #2:

Thank you very much for these constructive comments. We have revised the manuscript carefully based on these suggestions. The following are our point-by-point replies.

The manuscript by Haijun Yang et al. offers a comprehensive description of the physical mechanisms influenced by the presence / absence of large landform systems. The article is commendably clear and well-written. The responses to the reviewers' inquiries provide valuable insights that enhance our understanding of the authors' methodology.

The authors appropriately acknowledge the study's limitations in the conclusion, notably highlighting factors such as the paleogeography anchored in the present and the absence of atmospheric CO₂ variation, preventing direct comparison with the evolution of Cenozoic climates. I fully concur with this assessment.

However, in the conclusion, certain sentences imply that Tibet played a dominant role in the evolution of deep-water formation. To enhance clarity, it is advisable to ensure that the key points and the summary explicitly convey the nuanced nature of the study's findings and the limitations acknowledged in the conclusion. This adjustment will contribute to a more precise and comprehensive understanding of the research presented.

Responses: Thank you very much for these suggestions.

In Section summary and discussion, we added statement in lines 248-253 of the revised text to explicitly address the goal of this study as “This research marks the beginning of our investigation into the effects of the presence or absence of various major mountain ranges on the GMOC, as well as their cumulative impact under modern conditions. Our research aims to specifically discern the impact of topography on ocean circulation from other contributing factors, by incorporating modern bathymetry and continental configurations, current greenhouse gas levels, incident solar radiation, and orbital parameters in our experimental setup.”

And we also added statements to address the limitation of this study. In lines 272-276 of the text we state that “The setup of modern land-sea mask and ocean gateways in our experiments implies that the TP impact on the GMOC could be exaggerated. After all, the Tethys Seaway closing, the Panama Isthmus closing and the Bering Strait opening all occurred after the TP uplift⁸, all of which may lead to significant increases in the continental ice sheets in both hemispheres and enhance the NADW formation⁵¹⁻⁵³.” Further in lines 280-282 we state that “The TP uplift may have resulted in a decrease of atmospheric CO₂ level, which can also enhance the NADW formation due to the growth of large continental ice sheets in the NH⁴⁹.”

1. Key 1 : The authors contrast their results with previous studies, but the conditions of the Jones and Cecci (2017) study are different, so the comparison seems risky to me.

Responses: Thank you very much for this comment.

In lines 18-25 of the revised text, we listed three reasons that cause the different overturning modes between the Atlantic and Pacific: the asymmetry of net surface freshwater flux, the basin geometry and the ocean gateways. The key point #1 is also revised.

Jones and Cecci (2017)'s study focused on the effect of ocean basin geometry on the GMOC. They showed that the North Atlantic has higher salinity than the North Pacific because the Atlantic basin is narrower than the Pacific basin. They further explained that, because the southward western boundary current associated with the wind-driven subpolar gyre has higher velocity in the wide basin than in the narrow basin, it overwhelms the northward western boundary current associated with the MOC for wide-basin sinking, so freshwater is brought from the far north of the domain southward and forms a pool on the western boundary in the wide basin. The fresh pool suppresses local convection and spreads eastward, leading to low salinities in the north of the wide basin for wide-basin sinking. While in the narrow basin, the northward MOC western boundary current overcomes the southward western boundary current associated with the wind-driven subpolar gyre, bringing salty water from lower latitudes northward and enabling deep-water mass formation.

Jobes and Cecci (2017) did a perfect work to understand the different overturning modes from the point of view of ocean dynamics, particularly the wind-driven circulation dynamics, through sensitivity experiments using an ocean model (MITgcm). In their work, readers can learn a lot about how the wind-driven circulation affects the thermohaline circulation and their interplay.

Our work focused on the effect of the surface freshwater flux on the GMOC under the condition of current realistic ocean geometry. We just would like to show that, provided with an ocean geometry, the continental orography may significantly affect the global hydrological cycle and thus the surface freshwater fluxed over different oceans, which can cause remarkable changes of overturning modes in different oceans.

Our conclusion does not have to contradict to that of Jones and Cecci (2017), neither we compare our results with that of Jones and Cecci (2017). Our results are obtained from a coupled Earth system model while Jones and Cecci's results were obtained from an ocean GCM.

In lines 18-25 of the revised text, this paragraph is rewritten as follows.

“The asymmetry of net surface freshwater flux is often cited as the cause of different overturning modes between the Atlantic and Pacific. The North Atlantic has higher sea-surface salinity (SSS) than the North Pacific because the former is a net evaporation basin, while the latter is nearly neutral⁹. Additionally, ocean basin geometry plays a role in the different overturning modes. Research indicated that narrow basins are more conducive to deep overturning circulation than wide basins^{2,10,11}. Furthermore, ocean gateways also contribute to the different overturning modes. The opening of the Drake Passage/Tasman Seaway in the late Eocene is thought to have promoted the NADW formation and thus the AMOC formation¹².”

2. *L84: add more recent references*

Responses: Thank you very much for this suggestion. We have replaced ref. 12 with a more relevant reference “Hutchinson, D. K., Coxall, H. K., O’Regan, M. *et al.* Arctic closure as a trigger for Atlantic overturning at the Eocene-Oligocene Transition. *Nat Commun* **10**, 3797 (2019)”. This is also suggested by Reviewer #1.

3. *L89: The authors are encouraged to update the age of glacial inception in Antarctica, recognizing that the onset of glaciation in Antarctica is typically dated to the Late Eocene. This timing is attributed to a CO₂ atmospheric level that was considered too high before the Late Eocene. Please consider incorporating a relevant reference to support this updated information, providing a solid foundation for the revised age of glacial inception in Antarctica.*

Responses: Thank you very much for this comment.

In lines 29-32 of the revised text, this statement is rewritten as follows.

“Although the transantarctic and Gamburtsev Mountains over Antarctica were likely already present at the start of the Cenozoic (65 Ma)^{18,19}, the first large-scale glaciation of Antarctica is believed to have occurred during the Eocene-Oligocene transition period (34-33.5 Ma)^{20,21}”

We added two relevant references as ref. 20, 21 in the revised manuscript:

20. Jamieson, S. S. R., Sugden, D. E., & Hulton, N. R. J. The evolution of the sub-glacial landscape of Antarctica. *Earth and Planetary Science Letters* **293**, 1-27 (2010).

21. Jamieson, S. S. R., Ross, N., Paxman, G. J. G., Clubb, F. J. An ancient river landscape preserved beneath the East Antarctic Ice Sheet. *Nat Commun* **14**, 6507 (2023).

4. *L90: Change the reference.*

Responses: Thank you very much for this suggestion. The old reference 20 is changed to Boschmann (2021) (ref. 22 in the revised manuscript). And the statement about AMs is changed to “The uplift of Andes Mountains (AMs) started in the Late Cretaceous (~70 Ma)²² and matured around 15-10 Ma^{23,24}.”

5. *L83-99: The paragraph in question lacks clarity as the authors attempt to draw a comparison between the age of Antarctic glaciation and the age of the Andes uplift. It is recommended that the authors revisit and revise this paragraph to articulate the comparison more explicitly, providing a clearer and more coherent presentation of the relationship between the two geological events.*

Responses: Thank you very much for this comment. This paragraph is revised significantly following suggestions from you and Reviewer#1. The references within are also changed and updated. The revised paragraph reads as follows:

“Geological evidence also suggests that the uplift of large continental mountains has had a significant impact on the climate¹³. The evolution of continental terrain holds the potential to trigger large-scale transitions in the global MOC (GMOC)¹⁴⁻¹⁶. The Rocky Mountains (RMs) rose from the sea level about 80 Ma¹⁷, and reached its current elevation about 45 Ma⁸. Although the transantarctic and Gamburtsev Mountains over Antarctica were likely already present at the start of the Cenozoic (65 Ma)^{18,19}, the first large-scale glaciation of Antarctica is believed to have occurred during the Eocene-Oligocene transition period (34-33.5 Ma)^{20,21}. The uplift of Andes Mountains (AMs) started in the Late Cretaceous (~70 Ma)²² and matured around 15-10 Ma^{23,24}. The uplift of these mountains predated the onset of the NADW formation. The Greenland (GL) underwent its initial phase of uplift in the late Miocene (11-10 Ma)²⁵. The timeline for the formation of the Tibetan Plateau (TP) is a topic of highly debate. Some studies argue that parts of the TP were already in place during the late Eocene (38-33 Ma)^{26,27}, while other research suggest that the TP's rapid and main uplift occurred between 10 and 8 Ma²⁸⁻³¹. A more recent study proposes that most of the TP had attained its current elevation before the Mid-Miocene (15 Ma)³². This timing coincides with the onset of NADW formation, suggesting a possible link between the TP uplift and the development of NADW. Recent research also indicates that the TP is a critical factor affecting changes in the GMOC^{33,34}. Nevertheless, it is important to recognize that the chronology of the uplift of major mountain ranges remains a subject of intense discussion and investigation.”

6. *Leier et al. 2013: First uplift Early Miocene, second uplift Late Miocene – Andes Bolivia*

Responses: Thank you very much for this suggestion. We added the newer study of Leier et al. (2013) as the new reference 24.

7. *L142-151: The manuscript does not thoroughly discuss the intensification of the Pacific Meridional Overturning Circulation (PMOC) in relation to the uplift of the Andes*

Responses: Thank you very much for this comment. In this study, we did not provide detailed analyses on the PMOC change in response to the uplift of the Andes mountains (AMs), which, we think, deserve a full-length paper. We are working intensely on this issue through more sensitivity experiments. On page 2 of the revised manuscript, we delete the key point #5.

In lines 188-192 of the revised text, we briefly discussed the mechanism of the PMOC intensification in response to the uplift of the AMs as follows.

“The presence of the AM reduces the equatorial trade wind, but amplifies the off-equatorial Ekman pumping (Extended Data Fig. 5a), thereby boosting the wind-driven STC in the South Indo-Pacific (Extended Data Fig. 5b) and augmenting the thermohaline component of the PMOC there, leading to a stronger PMOC (Fig. 3e2).”

8. *Figure 4 -Line 8: Does the case Flat2Real correspond to the end of the experiment when all reliefs are uplifted (year 5601-6000)?*

Responses: Thank you very much for this comment. Yes, the equilibrium change in Flat2Real with respect to Flat corresponds to year 5601-6000 of Flat2Real.

9. *L201: During the summer in Antarctica, the flat topography of the continent is likely to promote convection. Additionally, the presence of a high-pressure anomaly can indeed be seasonal.*

Responses: Thank you very much for this comment. We agree that “the flat topography in Antarctica during the summer is likely to promote convection”, however, this convection is much weaker than that in the presence of the AT mountains. It is also likely that “the presence of a high-pressure anomaly can be seasonal”. We would like to say that on seasonal timescale, the effect of convection or high-pressure anomaly cannot lead to significant change in AABW and AMOC/PMOC, which has to rely on the accumulated annual changes.

10. *Figure 5a1: the figure 5a1 represents the difference Real minus Flat (on the plot) or Flat2Real minus flat (in the caption)*

Responses: Thank you very much for this comment. Fig. 5a1 represents the difference between Flat2Real (year 5601-6000) and Flat. which is practically identical to the difference between Real and Flat. We corrected the small mistake on the plot.

11. L246-262: The manuscript describes the retreat of sea ice in the subpolar Atlantic, but the underlying cause of this retreat is not clearly explained. It is crucial to address this gap by providing a detailed and explicit discussion on the factors driving the retreat of sea ice in the subpolar Atlantic.

Responses: Thank you very much for this comment.

This paragraph is revised and states explicitly what initialize the sea ice change in the subpolar Atlantic (see below). Please also refer to lines 195-221 of the revised text.

“The AMOC gradually increases in the first few hundred years after the TP uplift in both Flat2Real and TP+AT, followed by an acceleration and eventual return to a normal state in Real (Figs. 1, 2b). The latter process is accompanied by a swift sea-ice loss in the subpolar Atlantic. In response to the TP uplift, more atmospheric moisture converges (diverges) over the North Pacific (Atlantic) (Extended Data Figs. 2b2, 2b5, 3a2, 3a3), shutting down the PMOC and triggering a gradual increase of the AMOC at the same time. The latter enhances the northward heat transport in the Atlantic, leading to a gradual retreat of sea ice in the subpolar Atlantic. The sea ice retreat is also helped by the anomalous northward Ekman flow, forced by the anomalous easterlies over the subpolar Atlantic (Extended Data Fig. 3b2).

The evolution of sea ice in Flat2Real is shown in Fig. 6. The northward retreat of sea ice is illustrated by the sea ice velocity in Fig. 6b, which leads to additional freshwater loss in this region (Extended Data Figs. 2b2, 2b5). This freshwater loss in turn increases SSD, the NADW formation and thus the AMOC consequently. In the first few hundred years after adding the TP, the sea-ice in the subpolar Atlantic retreats northward slightly (Fig. 6b), and the March mixed layer depth (MLD) deepens slightly (Fig. 6a), corresponding to a gradual increase of the AMOC. About 500 years after the TP uplift, the collective effects of accumulated saline water in the subpolar Atlantic and Ekman pumping in Southern Ocean accelerate the AMOC, so that the sea-ice margin retreats rapidly northward (Fig. 6b, dashed red curve), resulting in a large amount of freshwater flux loss in the subpolar Atlantic, a rapid deepening of the MLD (Fig. 6a), and a further enhancement of the AMOC (Figs. 1, 2b). The sea-ice margin reaches its quasi-equilibrium roughly 1000 years after the TP uplift (Fig. 6b), accompanied by significant sea-ice melting in the GIN seas. The sea-ice evolution in TP+AT (Extended Data Figs. 7a-b) displays a similar pattern to that in Flat2Real, while changes in sea ice and MLD in TP are minimal (Extended Data Figs. 7c-d). The evolutions of AMOC, the MLD and sea-ice coverage and margin in Flat2Real and TP+AT suggest a positive feedback between the AMOC and sea ice changes, which eventually leads to the establishment of AMOC. This feedback has been shown in many previous studies (e.g., Brady and Otto-Bliesner 2011; Yang and Wen 2020).”

Reference:

47. Brady, E. C., and B. L. Otto-Bliesner, 2011: The role of meltwater induced subsurface ocean warming in regulating the Atlantic meridional overturning in glacial climate simulations. *Climate Dyn.*, 37, 1517–1532, <https://doi.org/10.1007/s00382-010-0925-9>

12. L289-291: the authors explained clearly on lines 289-291 that this study is based on sensitivity experiments (presence or absence of relief) and does not mimic past climate evolution. I fully agree with that. I regret that the key points and the abstract are less explicit.

Responses: Thank you very much for this comment.

In the revised manuscript, on page 2 the key point #1 is changed to “A series of coupled model sensitivity experiments with/without continental orography suggest that the thermohaline circulation could occur either in the Pacific or in the Atlantic, depending much on how the continental giant mountains affect the pattern of global hydrological cycle.”, which explicitly state that all results are based on sensitivity experiments.

In abstract, we explicitly state that “we design a series of coupled model *sensitivity* experiments to investigate ...”, and tone down the TP’s effect by removing the word “*paramount*”.

13. Then the authors compare with their results with the climate evolution during the Cenozoic. I disagree with the sentence (L300) suggesting that “this study establishes the direction of MOC change as a function of increasing uplift over time”. The position of continents (and relief) remains unchanged, as are the gateways (these limitations are indicated on L307).

Responses: Thank you very much for this comment. In the revised manuscript, this paragraph is revised significantly. Please refer to lines 248-264 of the revised text.

Particularly, the last sentence of this paragraph is changed to “By using experiment Flat as a starting point and tracing the changes due to various uplifts, we can quantify the linkages between uplifts and climate changes and understand how the MOC changes as progressive uplift of continental mountains.”

14. The clarification provided in lines 289-291 regarding the study's reliance on sensitivity experiments rather than a direct mimicry of past climate evolution is appreciated. However,

there seems to be a discrepancy in the key points and abstract, where the explicitness might be lacking.

Responses: Thank you very much for this comment. Please refer to our replies to Q.12-13, the key points, the abstract and the old paragraph in lines 289-300 are rewritten in the revised manuscript, in which both the consistency and explicitness are improved significantly.

15. Concerning the comparison with the climate evolution during the Cenozoic, there is disagreement with the sentence on line 300 suggesting that " this study establishes the direction of MOC change as a function of increasing uplift over time ". The disagreement arises from the understanding that the positions of continents (including relief) and gateways remain unchanged, as stated on line 307, and this should be reflected in the conclusions. A careful review and adjustment of these statements in the key points and abstract will contribute to a more accurate representation of the study's findings and limitations.

Responses: Thank you very much for these suggestions. Please also refer to our replies to Q.12-14, the key points, the abstract and Section "Summary and discussion" are rewritten to better state the findings and limitations of this study.

REVIEWERS' COMMENTS

Reviewer #1 (Remarks to the Author):

This is the third review of “Which Mountains Matter to Global Meridional Overturning Circulation” by Haijun Yang and coauthors for potential publication in Nature Communications.

I am happy with the responses to the comments and the corrections made by the authors. In my opinion, the manuscript is now suitable for publication.

I still give a couple of wording suggestions below.

I am also wondering whether the model outputs used in the manuscript will be made publicly accessible to comply with latest standards in climate modelling and to promote open science.

l. 66-71. “and likely began to develop at a later stage. Consequently, the modern AMOC might initially develop in the late Miocene (about 12-9 Ma) and not be fully established until the late Pliocene to early Pleistocene (about 4-3 Ma). Nonetheless, a recent study from DeepMIP project found that neither model results nor proxy data suggest NADW formation during the early Eocene, while the evidence for NPDW formation remains inconclusive.”

=> and likely began to develop at a later stage. However, a recent study from the DeepMIP project found that neither model results nor proxy data suggest NADW formation during the early Eocene, while evidence for NPDW formation remains elusive. Consequently, the modern AMOC might initially develop in the late Miocene (about 12-9 Ma) and not be fully established until the late Pliocene to early Pleistocene (about 4-3 Ma).

l. 71. “AMOC and PMOC remains a topic” => AMOC and PMOC therefore remains a topic

l. 83. “large-scale transitions in the global MOC (GMOC).” => large-scale transitions in the global MOC (GMOC), and in the following, we briefly review the uplift periods of the major modern mountain ranges.

l. 303. “This research marks the beginning of our investigation into the effects of the presence or absence...” => The results presented here are a first step toward a better understanding of the coupled impacts of the presence or absence...

l. 305. “Our research aims to specifically discern” => Our experiments are specifically designed to disentangle

l. 306. “by incorporating [...] in our experimental setup” => thereby keeping the bathymetry and continental configuration, as well as greenhouse gases, incident solar radiation and orbital parameters, to their modern state in our experimental setup.

l. 309. “explore the linkages between uplift regions and climate changes in remote areas of the modern

world". Consider changing to: explore the links and teleconnections between uplift regions and climate?

I. 310. "The sequential mountain uplift experiment Flat2Real provides insight into specific periods of paleoclimate, such as the climate development in the late Cenozoic." This sentence reads weirdly.

Consider changing to: Moreover, the sequential mountain uplift experiment Flat2Real may offer interesting insights about the long-term impact of mountains on Cenozoic climate change, notwithstanding significant caveats.

Then I suggest that you start a new paragraph with the next sentence ("Previous studies suggested that the TP uplift played a key role in shaping Cenozoic climate...")

I. 320-321. If you follow the comment just above, you can remove the following: "It should be noted [...] the credibility of model's results" because you already mentioned that there are caveats (or limitations). Just change "For instance" with "However" on line 322 and start the paragraph with it.

I. 328. Remove "After all".

I. 330. "lead to significant increases in continental ice sheets in both hemispheres and enhance the NADW formation" => lead to significant changes in ocean circulation and modulate NADW formation.

I. 335-336. Why only the TP uplift and not that of the other mountain ranges discussed?

I. 341. "We want to emphasize" => We still want to emphasize.

I. 345-347. "of debate. Understanding which mountains exert control over the MOC is critical for comprehending oceans' roles in past and future climate transitions. We hope that this study will help clarify this issue and reduce the controversy surrounding it."

=> of debate, and our results contribute to a better understanding of the controls exerted by mountain ranges over the MOC, which is critical for a better comprehension of the role of the ocean in past and future climate transitions.

Reviewer #2 (Remarks to the Author):

The authors have diligently incorporated the reviewers' comments to refine the manuscript. From my perspective, the approach utilized demonstrates that the presence or absence of a Tibetan plateau influences the Atlantic Meridional Overturning Circulation (AMOC) and Pacific Meridional Overturning Circulation (PMOC) in these sensitivity experiments. However, it appears challenging to extrapolate these findings to the actual evolution of the AMOC in the Cenozoic.

The uplift of landforms significantly shapes Earth's landscape over tens of millions of years, with major temporal overlaps that diverge from the events discussed here. Additionally, factors such as changes in ocean passages, variations in atmospheric carbon dioxide (pCO₂) levels, and the glaciation of Antarctica and Greenland also play pivotal roles in Earth's climate evolution.

While the study provides valuable insights into the analyzed mechanisms, it does not conclusively establish that the presence of the Tibetan plateau (along with Antarctica topography) is the sole triggering factor for the AMOC and the demise of the PMOC in the Cenozoic.

L28: I find the wording of the key point 1 to be questionable as it refers to the thermohaline circulation in general, whereas it should specifically address the location of convection, which ultimately impacts the Atlantic Meridional Overturning Circulation (AMOC) and Pacific Meridional Overturning Circulation (PMOC)."

L42: The presence of Greenland's topography is unsurprising. Convection in the North Atlantic Ocean is intensified by the Greenland ice sheet, with surface albedo playing a significant role. (Pillot et al. 2022).

L39: Key points should focus on the main results of the work itself and not reference other papers from the literature. (ref 1 and 14 called in key point 4)

L62 : The authors assert that geological evidence elucidates the location of deep water formation. What evidences ? add a reference.

L79 : It refers more to the paper by Toggweiler, J. R. & Bjornsson, H. Drake passage and palaeoclimate. J. Quat. Sci. 15, 319–328 (2000) as indicated in Hutchinson et al. 2019

L83-84: The authors can cite the paper by Chamberlain et al. (2012) to discuss the evolution of the Rocky Mountains. According to their findings, the current shape of the Rocky Mountains was established during the Oligocene. However, it is worth noting that some landforms within the range have undergone collapse since that time.

L85 : add a capital to Transantarctic Mountain. It appears that the authors did not include Antarctic landforms in their simulations. They incorporate the modern elevation of the Antarctic ice sheet into their runs (after the Fig 1 in SI).

L89: "The uplift of these mountains predated the onset of the NADW formation". The authors should recall that the simulated Meridional Overturning Circulation (MOC) responds to the absence or presence of mountain ranges. While it is agreed that the Rocky Mountains were largely uplifted before the onset of the North Atlantic Deep Water (NADW) formation, the situation is more debatable for the Andes. The uplift of the Andes may not have occurred linearly over time, and it is possible that the largest uplift took place during the Late Cenozoic.

L105: The added Antarctic (AT) topography during the simulation primarily represents that of the Antarctic ice sheet, rather than the landforms (Transantarctic and Gamburtsev Mountains). This also applies to the Greenland (GL) ice sheet. As explained in the "Methods" section, the continental ice sheets are treated as bright rocks with an albedo similar to that of a bare soil. This contrasts with the "REAL" case, where the albedo of the surface should be realistic. It would be intriguing to observe the Pacific Meridional Overturning Circulation (PMOC) and Atlantic Meridional Overturning Circulation (AMOC) when the realistic albedo of the surface is applied in the "REAL" case. It is surprising to note that

the intensity of the AMOC reaches 18 Sv without considering the albedo of the Antarctic and Greenland ice sheets, as observed in the preindustrial control run with ice sheets conducted by Shields et al. (2012), although it should be noted that the models may not be exactly the same.

L134 : “irrespective of the order in which they were introduced to Flat”. Thus the authors have tested different scenarios (not shown in extended data ?)

L288 : It is indeed challenging to draw conclusions about the role of the Tibetan Plateau (TP) on modern-day thermohaline circulation. The abrupt addition of the TP in the experiment is far from realistic, as acknowledged by the authors. These are sensitivity experiments designed to explore potential impacts rather than replicate real-world conditions.

Moreover, it's important to note that the uplift of the TP did not commence at 8-10 million years ago (Ma). Evidence suggests that at least parts of the Tibetan plateau, such as South and Central Tibet, were already elevated during the Mid-Miocene, thus predating the establishment of a modern Atlantic Meridional Overturning Circulation (AMOC).

L481 : Shields et al => ref 55 not 56

Replies to Reviewer #1:

Thank you very much for invaluable suggestions to this manuscript.

This is the third review of “Which Mountains Matter to Global Meridional Overturning Circulation” by Haijun Yang and coauthors for potential publication in Nature Communications.

I am happy with the responses to the comments and the corrections made by the authors. In my opinion, the manuscript is now suitable for publication.

I still give a couple of wording suggestions below.

I am also wondering whether the model outputs used in the manuscript will be made publicly accessible to comply with latest standards in climate modelling and to promote open science.

Responses: Yes. All model results in this study are available from the corresponding authors upon request. We also add Data availability in the article.

1. l. 66-71. *“and likely began to develop at a later stage. Consequently, the modern AMOC might initially develop in the late Miocene (about 12-9 Ma) and not be fully established until the late Pliocene to early Pleistocene (about 4-3 Ma). Nonetheless, a recent study from DeepMIP project found that neither model results nor proxy data suggest NADW formation during the early Eocene, while the evidence for NPDW formation remains inconclusive.” => and likely began to develop at a later stage. However, a recent study from the DeepMIP project found that neither model results nor proxy data suggest NADW formation during the early Eocene, while evidence for NPDW formation remains elusive. Consequently, the modern AMOC might initially develop in the late Miocene (about 12-9 Ma) and not be fully established until the late Pliocene to early Pleistocene (about 4-3 Ma).* **Revised.**
2. l. 71. *“AMOC and PMOC remains a topic ” => AMOC and PMOC therefore remains a topic* **Revised.**
3. l. 83. *“large-scale transitions in the global MOC (GMOC). ” => large-scale transitions in the global MOC (GMOC), and in the following, we briefly review the uplift periods of the major modern mountain ranges.* **Revised.**
4. l. 303. *“This research marks the beginning of our investigation into the effects of the presence or absence …” => The results presented here are a first step toward a better understanding of the coupled impacts of the presence or absence …* **Revised.**

5. l. 305. *“Our research aims to specifically discern ” => Our experiments are specifically designed to disentangle* **Revised.**
6. l. 306. *“by incorporating [···] in our experimental setup ” => thereby keeping the bathymetry and continental configuration, as well as greenhouse gases, incident solar radiation and orbital parameters, to their modern state in our experimental setup.* **Revised.**
7. l. 309. *“explore the linkages between uplift regions and climate changes in remote areas of the modern world ” . Consider changing to: explore the links and teleconnections between uplift regions and climate?* **Revised.**
8. l. 310. *“The sequential mountain uplift experiment Flat2Real provides insight into specific periods of paleoclimate, such as the climate development in the late Cenozoic. ” This sentence reads weirdly. Consider changing to: Moreover, the sequential mountain uplift experiment Flat2Real may offer interesting insights about the long-term impact of mountains on Cenozoic climate change, notwithstanding significant caveats. Then I suggest that you start a new paragraph with the next sentence (“Previous studies suggested that the TP uplift played a key role in shaping Cenozoic climate··· ”)*

Responses: Thank you very much. We have revised the sentences according to your suggestions.

9. l. 320-321. *If you follow the comment just above, you can remove the following: “It should be noted [···] the credibility of model's results ” because you already mentioned that there are caveats (or limitations). Just change “For instance ” with “However ” on line 322 and start the paragraph with it.*

Responses: Thank you very much for this comment. We have revised the sentences according to your suggestions.

10. l. 328. *Remove “After all ” .* **Revised.**
11. l. 330. *“lead to significant increases in continental ice sheets in both hemispheres and enhance the NADW formation ” => lead to significant changes in ocean circulation and modulate NADW formation.* **Revised.**

12. l. 335-336. *Why only the TP uplift and not that of the other mountain ranges discussed?*

Responses: Thank you for your comments. I guess other mountains' uplift have the same effect on the atmospheric CO₂. The TP's effect, not other mountains, was explicitly discussed in Ref. 49. So here it is safer to just mention the TP's effect on CO₂.

13. l. 341. *“We want to emphasize” => We still want to emphasize.* **Revised.**

14. l. 345-347. *“of debate. Understanding which mountains exert control over the MOC is critical for comprehending oceans’ roles in past and future climate transitions. We hope that this study will help clarify this issue and reduce the controversy surrounding it.” => of debate, and our results contribute to a better understanding of the controls exerted by mountain ranges over the MOC, which is critical for a better comprehension of the role of the ocean in past and future climate transitions.* **Revised.**

Replies to Reviewer #2:

Thank you very much for invaluable comments to this manuscript, which help to improve the manuscript tremendously.

The authors have diligently incorporated the reviewers' comments to refine the manuscript. From my perspective, the approach utilized demonstrates that the presence or absence of a Tibetan plateau influences the Atlantic Meridional Overturning Circulation (AMOC) and Pacific Meridional Overturning Circulation (PMOC) in these sensitivity experiments. However, it appears challenging to extrapolate these findings to the actual evolution of the AMOC in the Cenozoic.

The uplift of landforms significantly shapes Earth's landscape over tens of millions of years, with major temporal overlaps that diverge from the events discussed here. Additionally, factors such as changes in ocean passages, variations in atmospheric carbon dioxide (pCO₂) levels, and the glaciation of Antarctica and Greenland also play pivotal roles in Earth's climate evolution.

While the study provides valuable insights into the analyzed mechanisms, it does not conclusively establish that the presence of the Tibetan plateau (along with Antarctica topography) is the sole triggering factor for the AMOC and the demise of the PMOC in the Cenozoic.

Responses: Thank you very much for the comments. We agree your assessment on this work. Actually, in Section summary and discussion, we have discussed the problems you concerned very carefully in three paragraphs.

First, we state explicitly that “*The results presented here are a first step toward a better understanding of the coupled impacts of the presence or absence of various major mountain ranges on the GMOC ...*” There are lots of other factors that should be considered if we want to reach a fully understanding of the climate evolution in the Cenozoic. There is a very long way to go.

1. *L28: I find the wording of the key point 1 to be questionable as it refers to the thermohaline circulation in general, whereas it should specifically address the location of convection, which ultimately impacts the Atlantic Meridional Overturning Circulation (AMOC) and Pacific Meridional Overturning Circulation (PMOC).*”

Responses: Thank you for your suggestions. We have changed key point 1 to “A series of coupled model sensitivity experiments with/without continental orography suggest that the *deep-water formation* could occur either in the Pacific or in the Atlantic, depending much on how the continental giant mountains affect the pattern of global hydrological cycle.”

2. *L42: The presence of Greenland's topography is unsurprising. Convection in the North Atlantic Ocean is intensified by the Greenland ice sheet, with surface albedo playing a significant role. (Pillot et al. 2022).*

Responses: Thank you very much for this comment. In this study, we just explored the effect of the Greenland topography, and did not study the effect of the Greenland ice sheet. The albedo effect related to the Greenland ice sheet is actually a very important issue that we are going to study in the near future.

3. *L39: Key points should focus on the main results of the work itself and not reference other papers from the literature. (ref 1 and 14 called in key point 4)*

Responses: Thank you very much. We have changed key point 4 to “*The Rocky Mountains have no significant effect on either the PMOC or the AMOC.*”

4. *L62: The authors assert that geological evidence elucidates the location of deep water formation. What evidences? add a reference.*

Responses: Thank you very much for the comment. The references 4-7 is labeled in the following sentences in the same paragraph.

5. *L79: It refers more to the paper by Toggweiler, J. R. & Bjornsson, H. Drake passage and palaeoclimate. J. Quat. Sci. 15, 319 – 328 (2000) as indicated in Hutchinson et al. 2019*

Responses: Thank you very much for this suggestion. We have changed the ref. 12 in this sentence to “Toggweiler, J. R. & Bjornsson, H. Drake passage and palaeoclimate. J. Quat. Sci. 15, 319 – 328 (2000)”.

6. *L83-84: The authors can cite the paper by Chamberlain et al. (2012) to discuss the evolution of the Rocky Mountains. According to their findings, the current shape of the Rocky Mountains was established during the Oligocene. However, it is worth noting that some landforms within the range have undergone collapse since that time.*

Responses: Thank you very much for this suggestion. We have replaced the old ref. 17 with Chamberlain et al. (2012).

7. L85: *add a capital to Transantarctic Mountain. It appears that the authors did not include Antarctic landforms in their simulations. They incorporate the modern elevation of the Antarctic ice sheet into their runs (after the Fig 1 in SI).*

Responses: Thank you for your suggestion. We have added capital to “*Transantarctic Mountain*”. In this study, we did not include Antarctic landforms in our simulations.

8. L89: *“The uplift of these mountains predated the onset of the NADW formation ” . The authors should recall that the simulated Meridional Overturning Circulation (MOC) responds to the absence or presence of mountain ranges. While it is agreed that the Rocky Mountains were largely uplifted before the onset of the North Atlantic Deep Water (NADW) formation, the situation is more debatable for the Andes. The uplift of the Andes may not have occurred linearly over time, and it is possible that the largest uplift took place during the Late Cenozoic.*

Responses: Thank you very much for the invaluable comments. We also found that the effect of the Andes mountains on the MOC is not that obvious. We are working intensely on this issue using a high-resolution version of CESM2.0. Since the Andes mountains is narrow in the east-west direction, the model resolution might be an important issue in simulating its effect on global climate. We also made discussion on this in section *Summary and discussion*.

9. L105: *The added Antarctic (AT) topography during the simulation primarily represents that of the Antarctic ice sheet, rather than the landforms (Transantarctic and Gamburtsev Mountains). This also applies to the Greenland (GL) ice sheet. As explained in the "Methods" section, the continental ice sheets are treated as bright rocks with an albedo similar to that of a bare soil. This contrasts with the "REAL" case, where the albedo of the surface should be realistic. It would be intriguing to observe the Pacific Meridional Overturning Circulation (PMOC) and Atlantic Meridional Overturning Circulation (AMOC) when the realistic albedo of the surface is applied in the "REAL" case. It is surprising to note that the intensity of the AMOC reaches 18 Sv without considering the albedo of the Antarctic and Greenland ice sheets, as observed in the preindustrial control run with ice sheets conducted by Shields et al. (2012), although it should be noted that the models may not be exactly the same.*

Responses: Thank you very much for the invaluable comments. Based on our simulations, we feel that the landforms of Antarctic, as well as the detailed albedo patterns appear to have very limited

effect on the mean strength of the PMOC and AMOC. This issue needs to be explored in-depth using latest version of coupled models. Actually, we are working intensely on the topographic climate effect using CESM2.1. It is a very time-consuming job and we may not have results until the late 2024.

10. L134: “irrespective of the order in which they were introduced to Flat ” . Thus the authors have tested different scenarios (not shown in extended data?)

Responses: Thank you for the comment. Yes, we tested lots of different scenarios, which were not shown in extended data. The basic concept is that the *equilibrium* response has to be independent on the order of the mountains added, otherwise the response in the final “Real” simulation will not be unique.

11. L288: It is indeed challenging to draw conclusions about the role of the Tibetan Plateau (TP) on modern-day thermohaline circulation. The abrupt addition of the TP in the experiment is far from realistic, as acknowledged by the authors. These are sensitivity experiments designed to explore potential impacts rather than replicate real-world conditions. Moreover, it's important to note that the uplift of the TP did not commence at 8-10 million years ago (Ma). Evidence suggests that at least parts of the Tibetan plateau, such as South and Central Tibet, were already elevated during the Mid-Miocene, thus predating the establishment of a modern Atlantic Meridional Overturning Circulation (AMOC).

Responses: Thank you very much for the comment. We agree that it is indeed challenging to draw conclusion about the role of the TP in modern-day thermocline circulation, given the challenges of accurately simulating the entire evolutionary process that occurred in reality. It's essential to recognize that among all the world's major mountain ranges, the TP holds the greatest potential to impact the AMOC. While we concur that the TP's uplift began well before 8-10 Ma, there is also evidence to suggest that its principal elevation occurred around this time frame.

This study represents an initial step towards understanding the significant influence of continental orography on Earth's climate formation. We are diligently advancing our research on the evolution of Earth's climate since the Cenozoic era through meticulously designed sensitivity experiments.

12. L481 : Shields et al => ref 55 not 56 **Revised.**